# Glucocorticoids paradoxically promote steroid resistance in B cell acute lymphoblastic leukemia through CXCR4/PLC signaling

Souleymane Abdoul-Azize [1] ✉, Rihab Hami[2], Gaetan Riou[1], Céline Derambure [3], Camille Charbonnier[3], Jean-Pierre Vannier[1], Monica L. Guzman [4], Pascale Schneider[1,5] & Olivier Boyer [1,6]

Glucocorticoid (GC) resistance in childhood relapsed B-cell acute lymphoblastic leukemia (B-ALL) represents an important challenge. Despite decades of clinical use, the mechanisms underlying resistance remain poorly understood. Here, we report that in B-ALL, GC paradoxically induce their own resistance by activating a phospholipase C (PLC)-mediated cell survival pathway through the chemokine receptor, CXCR4. We identify PLC as aberrantly activated in GC-resistant B-ALL and its inhibition is able to induce cell death by compromising several transcriptional programs. Mechanistically, dexamethasone (Dex) provokes CXCR4 signaling, resulting in the activation of PLC-dependent $Ca^{2+}$ and protein kinase C signaling pathways, which curtail anticancer activity. Treatment with a CXCR4 antagonist or a PLC inhibitor improves survival of Dex-treated NSG mice in vivo. CXCR4/PLC axis inhibition significantly reverses Dex resistance in B-ALL cell lines (in vitro and in vivo) and cells from Dex resistant ALL patients. Our study identifies how activation of the PLC signalosome in B-ALL by Dex limits the upfront efficacy of this chemotherapeutic agent.

Acute lymphoblastic leukemia (ALL) is the most common childhood cancer, representing approximately one-third of pediatric cancers and three-fourths of leukemias. The majority, 85–90%, is of B-lymphoblastic origin while 10–15% arise from T lymphoblasts[1–3]. Therapy of B-ALL was gradually adapted to the stratification of patients into different risk groups according to clinical and biological criteria. Although outcome for children with B-ALL has dramatically improved over the last decades, 20–25% children become resistant to chemotherapy[4]. The existence of this resistance from the start of primary treatment with glucocorticoids (GC)[5,6] indicates that GC

resistance is a main driver of refractoriness/relapse in B-ALL. Yet, the mechanistic basis of this primary GC resistance remain elusive. GC act via cytoplasmic GC receptors (GR) that translocate to the nucleus and then bind to target gene loci, inducing a transcriptional program that drives the expected apoptosis of leukemic cells. Yet, there is growing evidence that GC may also unexpectedly induce or facilitate intrinsic resistance in ALL[7,8]. For example, in T-ALL, GC elicit their own resistance in the presence of IL-7, by promoting upregulation of IL-7 receptor (IL-7R) expression, resulting in upregulation of the pro-survival protein BCL-2 through STAT5[7]. Although IL-7R is involved in

[1]Univ Rouen Normandie, Inserm, UMR 1234, F-76000 Rouen, France. [2]Univ Brest, Inserm, UMR 1101, F-29200 Brest, France. [3]Univ Rouen Normandie, Inserm, UMR 1245, Rouen, France. [4]Division of Hematology and Oncology, Department of Medicine, Weill Cornell Medicine, New York, NY, USA. [5]Rouen University Hospital, Department of Pediatric Immuno-Hemato-Oncology, F-76000 Rouen, France. [6]Rouen University Hospital, Department of Immunology and Biotherapy, F-76000 Rouen, France. ✉e-mail: souleymane.abdoul-azize@inserm.fr

the development of both B and T cells[7,9,10], it appears to be specific to T-ALL as a mechanism of resistance. In B-ALL in which, unlike T-ALL, IL-7R expression is repressed by GC[11], we previously evidenced a role for intracellular $Ca^{2+}$ signaling in the resistance to the anti-leukemic effects of GC. Indeed, by stimulating intracellular $Ca^{2+}$, GC trigger a pro-survival program in leukemic cells while intracellular $Ca^{2+}$ chelation improves GC sensitivity[8].

$Ca^{2+}$ ion controls several cellular functions through various signaling pathways. For example, aberrant $Ca^{2+}$ signaling has been demonstrated in cancers where it regulates cell proliferation, invasiveness, angiogenesis, migration, and metastasis[12–14]. Mutations in molecules controlling $Ca^{2+}$ homeostasis have been associated with an increased incidence of tumors[12–14], but many key elements of that association are not fully understood. In B cells, initial $Ca^{2+}$ signals are generated by the actions of cell surface receptors (CSR) through (i) engagement of the B cell receptor (BCR) by antigens or (ii) G-protein-coupled chemokine receptors[15–17]. Either way, CSR activate the phospholipase C (PLC) signaling pathway and the production of inositol 1, 4, 5-trisphosphate ($IP_3$) resulting in the release of $Ca^{2+}$ from the endoplasmic reticulum (ER) into the cytoplasm via $IP_3$ receptors ($IP_3R$) and diacylglycerol (DAG)-mediated activation of protein kinase C (PKC)[15]. This pathway plays a crucial role in many biological functions (e.g., proliferation and survival) by activating the transcription of various target genes[18]. Among the six PLC isoforms[18], PLCγ2 is a master regulator of B cell signaling, maturation, and function[10,18,19]. Accordingly, PLCγ dysfunction is associated with a variety of immune disorders and cancer[20,21]. For instance, in chronic lymphocytic leukemia, hypermorphic PLCγ2 mutations are associated with resistance to Bruton's tyrosine kinase (BTK) inhibition by ibrutinib, which provokes constitutive downstream signaling (e.g., $IP_3$ production and $Ca^{2+}$ release) and B cell proliferation[18,20,22,23]. In B-ALL, inhibition of calcineurin, with cyclosporin A increases caspase-3 activation in vitro and prolongs survival in vivo in combination with dasatinib in a mouse model of B-ALL[24]. Calcineurin is activated by $IP_3$-mediated $Ca^{2+}$ signaling, leading to translocation and nuclear activation of nuclear factor of activated T cells (NFAT)[25]. In B cell-like diffuse large B cell lymphoma (ABC DLBCL) cells, cyclosporin A or tacrolimus trigger potent cytotoxicity by repressing constitutive NFAT signaling[26]. These studies indicate that the PLCγ2 signaling pathway is an important regulator of B cell development but its role in B-ALL and drug resistance remains unknown.

GC directly affect the signaling of various surface receptors. A small hairpin RNA (shRNA) screen identified that BCR signaling (e.g., *BTK*) and early B cell development (e.g., *BCL6*) genes were repressed by GC in B-ALL[11]. Pre-BCR signaling plays a major role in clonal selection, proliferation, and subsequent maturation of pre-B cells[27,28]. Several studies have reported that pre-BCR is not functional in pre-B-ALL due to deletion, mutation, and splicing alterations[27,29]. For example, a study of 830 cases of pre-B-ALL from four clinical trials, shows that pre-BCR function was absent in 718 cases (87%)[29,30]. Pre-BCR has been suggested to function as a tumor suppressor due to the presence of non-functional *IgHM* gene rearrangements, thereby allowing leukemic pre-B cells to escape normal differentiation while avoiding the pre-BCR checkpoint[27]. Indeed, re-expression of a functional μ-chain in primary ALL cells with non-functional *IgHM* rearrangements[31] or successful reconstitution of pre-BCR signaling[32] effectively suppressed leukemic growth.

Besides the BCR, the other drivers of $Ca^{2+}$ signaling in B cells are G-protein-coupled chemokine receptors. Interestingly, C-X-C Motif Chemokine Receptor 4 (CXCR4), the receptor for SDF-1 induces PLCγ2 phosphorylation and activation in B cells[16]. SDF-1 and CXCR4 signaling exert growth-promoting and pro-survival effects in normal B cells and B-ALL cells[33,34]. Although activation of CXCR4 by GC has been reported in B-ALL[11], the signaling mechanism by which the CXCR4 pathway modulates GC sensitivity is unknown.

Here, we show that GC activate CXCR4/PLC signaling, an unwanted pro-survival pathway, that is associated with the early development of their own resistance, and that interfering with this pathway restores patients' leukemic cell killing and prolongs survival in mice.

## Results

### PLCγ2 is highly expressed and activated in B-ALL cells

$Ca^{2+}$ signaling in B cells is regulated by PLC among which PLCγ2 plays a key role in B lymphocyte development[10,19]. We first explored a dataset that is part of the Microarray Innovations In LEukemia (MILE) study program[35], which contains 1933 patients with acute and chronic leukemia, and analyzed overrepresented pathways using R2 database. Kyoto Encyclopedia of Genes and Genomes (KEGG) analysis revealed $Ca^{2+}$ signaling as the third most enriched pathway in this cohort[35] (Supplementary Fig. 1a). Comparing pathways overrepresented between leukemia ($n = 1933$) and non-leukemia/healthy ($n = 71$) patient samples, $Ca^{2+}$ signaling appeared as one of 13 pathways strongly enriched in leukemia (Supplementary Fig. 1b). When restricting the analysis to B-ALL only ($n = 80$) versus non-leukemia patient samples ($n = 71$), the $Ca^{2+}$ signaling pathway was consistently upregulated in B-ALL (Fig. 1a). Of all six PLC family members, PLCβ2, PLCβ3, and PLCγ2 play important roles in B cell signaling[10,18]. Because the function of these three PLCs in B-ALL is not clearly defined, we first compared their mRNA levels in a panel of 12 B-ALL cell lines[36]. As shown in Supplementary Fig. 1c, PLCγ2 mRNA level was highest in all cell lines compared to the other two PLCs (PLCβ2, PLCβ3). We then compared the expression of *PLCγ2* gene in human B-ALL to that in other cancer types[36]. mRNA levels of PLCγ2 were more highly expressed in B-ALL than in most other human cancers (ranked 4 of 40) (Fig. 1b and Supplementary Fig. 1d). Interestingly, PLCγ2 was detected mainly in B-ALL and weakly in non-leukemia patient samples (MILE dataset[35], Fig. 1c). Similar results were obtained when comparing *PLCγ2* expression of B-ALL to normal peripheral blood mononuclear cell (PBMC) samples by in silico analysis of public data[37] using the Oncomine database[38] (Fig. 1d and Supplementary Fig. 1e), or DLBCL compared to normal samples (GEPIA data[39], Supplementary Fig. 1f). These data suggest that the PLC pathway, and in particular PLCγ2, could play a role in B-ALL. As shown in Supplementary Fig. 1g, PLCγ2 hydrolyzes phosphatidylinositol-4,5-bisphosphate ($PtdIns(4,5)P_2$ or $PIP_2$) to generate two second messengers, $IP_3$ and DAG, resulting in the release of $Ca^{2+}$ from internal stores and PKC activation, respectively[40]. PKCβ is involved in BCR signaling and B cell survival[41]. Consistently, analysis of the same public data[37] showed increased mRNA levels of PLCγ2 targets such as ITPR3 and PRKCβ in B-ALL compared to normal PBMCs (Supplementary Fig. 1h–j). To further establish a direct connection between B-ALL and PLCγ2 signaling and activation, we analyzed the total level of PLCγ2 and its basal level of phosphorylation. PLCγ2 is phosphorylated on Tyr759 for enzyme activity. A basal phosphorylation of PLCγ2 was found in B-ALL cells (Fig. 1e), which was significantly higher in B-ALL cell lines compared to normal B cells (Fig. 1f). We also found a higher expression of total PLCγ2 (Fig. 1g) in the two B-ALL cell lines than in normal B cells. These data suggest that in B-ALL, PLC-γ2 could be aberrantly activated. To validate these results and assess the potential functionality of PLCγ2 in B-ALL, we performed $Ca^{2+}$ signaling experiments in the absence of extracellular $Ca^{2+}$, in order to mobilize only the ER $Ca^{2+}$ stores via the $IP_3$ receptor ($IP_3R$), which is a hallmark of PLCγ2 activation. Nalm-6 B-ALL cells were first treated with different PLC inhibitors, i.e., U73122, edelfosine, 3-Nitrocoumarin, SPK-601, D609, and manoalide, which all significantly decreased PLCγ2 constitutive activity, even at low concentration (Supplementary Fig. 2a, b), without affecting PLCγ2 protein levels (Supplementary Fig. 2c). $IP_3R$-mediated ER $Ca^{2+}$ release following carbachol (Cch) (Fig. 1h, j and Supplementary Fig. 2d) or ATP stimulation (Fig. 1i, k and Supplementary Fig. 2e) was also significantly reduced in presence of PLC inhibitors, suggesting that in B-ALL, PLC-γ2 is constitutively active and functional.

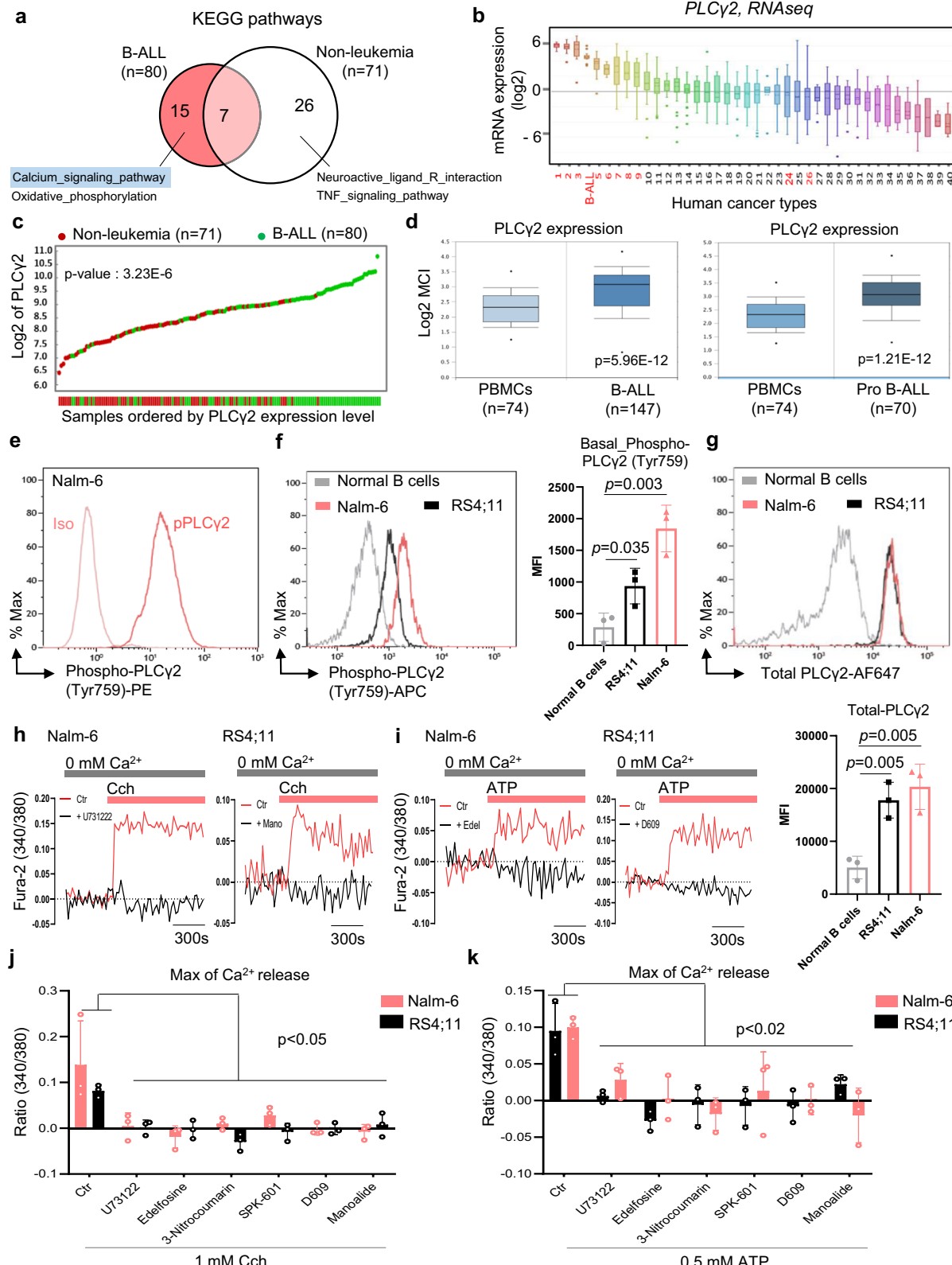

**PLCγ2 inhibitors decrease viability in B-ALL cells and Dexamethasone-resistant B-ALL cells**

To investigate whether the strong expression and activation of PLCγ2 observed in B-ALL affects cell survival and progression, cell death was evaluated in response to the inhibition of this basal activity. As shown in Fig. 2a, b, PLCγ2 inhibitors were able to significantly decrease cell viability in RS4;11 B-ALL cells, and Nalm-6 B-ALL cells,

respectively. An inhibitory effect was also observed in Reh B-ALL cells treated with U73122 (Supplementary Fig. 3a). U73122 showed anti-viability, even at low concentration, as measured by LIVE/DEAD® staining using flow cytometry in Nalm-6 B-ALL cells (Supplementary Fig. 3b). We then investigated which PLC downstream pathway affected cell viability. PLC controls the activation of several down-stream effectors, such as PKC activation and IP$_3$R-mediated Ca$^{2+}$

**Fig. 1 | PLCγ2 is highly expressed and active in B-ALL cells. a** Venn diagram of KEGG pathway analysis showing enriched pathways in non-leukemia and B-ALL samples from the R2 database in the Microarray Innovations in LEukemia (MILE) dataset ($n = 26$ enriched pathways in non-leukemia and $n = 15$ in B-ALL). Examples of enriched pathways are indicated for each subset. Two-tailed $t$-test, $p$-value < 0.05 for all pathways. **b** PLCγ2 mRNA ranked by expression level in B-ALL and other human tumors. Data were extracted from the cancer cell line encyclopedia (CCLE). Numbers refer to cancer cell types (see Supplementary Fig. 1d for listing) among which hematologic malignancies are shown in red. Boxplots show the mean, median, and 75th to 25th percentiles. **c** YY-graph of PLCγ2 mRNA expression levels in non-leukemia and B-ALL samples with expression levels ordered from left (low) to right (high) from (**a**). One-way analysis of variance (ANOVA) test. **d** PLCγ2 mRNA expression graphed as mean with SD between PBMCs and B-ALL samples in the Haferlach Leukemia dataset. Data were obtained from the Oncomine portal and shown as boxplots representing the 25th to 75th percentile range, mean, and median. Two-sided $t$-test. **e** An example of Nalm-6 cells staining with phospho-PLCγ2 antibody and isotype control. **f** Flow cytometry plots (left) and MFI (right) of the phosphorylated active form of PLCγ2 (phospho-PLCγ2 (Tyr759)). Data are mean ± SEM ($n = 3$ independent experiments). Two-tailed, unpaired Student's $t$-test. **g** Flow cytometry plots (left) and mean fluorescence intensity (MFI, right) of total PLCγ2. Data are mean ± SEM ($n = 3$ independent experiments). Two-tailed, unpaired Student's $t$-test. **h, i** Endoplasmic reticulum (ER) Ca$^{2+}$ release in Nalm-6 and RS4;11 cell lines stimulated with 1 mM carbachol, Cch (**h**) or 0.5 mM adenosine triphosphate, ATP (**i**). Cells were preincubated with the indicated PLC inhibitors during Fura-2 loading before stimulation. **j, k** Quantification of maximal ER Ca$^{2+}$ release from (**h, i**), respectively. Data are mean ± SEM ($n = 3$ independent experiments). Exact $P$-values from two-way ANOVA with Sidak's multiple comparisons test were as follows: **j** Nalm-6: Ctr vs. U73122, $p = 0.0001$; Ctr vs. Edelfosine, $p < 0.0001$; Ctr vs. 3-Nitrocoumarin, $p = 0.0002$; Ctr vs. SPK-601, $p = 0.0013$; Ctr vs. D609, $p < 0.0001$; Ctr vs. Manoalide, $p < 0.0001$. **j** RS4;11: Ctr vs. U73122, $p = 0.0296$; Ctr vs. Edelfosine, $p < 0.0186$; Ctr vs. 3-Nitrocoumarin, $p = 0.0013$; Ctr vs. SPK-601, $p = 0.0114$; Ctr vs. D609, $p = 0.0203$; Ctr vs. Manoalide, $p = 0.0484$. **k** Nalm-6: Ctr vs. U73122, $p = 0.0109$; Ctr vs. Edelfosine, $p = 0.0004$; Ctr vs. 3-Nitrocoumarin, $p = <0.0001$; Ctr vs. SPK-601, $p = 0.0017$; Ctr vs. D609, $p = 0.0003$; Ctr vs. Manoalide, $p < 0.0001$. **k** RS4;11: Ctr vs. U73122, $p = 0.0012$; Ctr vs. Edelfosine, $p < 0.0001$; Ctr vs. 3-Nitrocoumarin, $p = 0.0003$; Ctr vs. SPK-601, $p = 0.0002$; Ctr vs. D609, $p = 0.0002$; Ctr vs. Manoalide, $p = 0.0093$. Source data are provided as a Source Data file.

release from ER, leading to the activation of calcineurin/NFAT signaling[25]. As shown in Fig. 2c and Supplementary Fig. 3c, PKC inhibitors triggered cell death in Nalm-6 and RS4;11 cells. An inhibitory effect was also observed in both cell lines in the presence of 2-APB, an IP$_3$R-mediated Ca$^{2+}$ release inhibitor[42] (Fig. 2d) or calcineurin inhibitors (Fig. 2e and Supplementary Fig. 3d). To determine whether this induction of cell death was in part caused by an increase in apoptotic cell death, caspase-3 activity and annexin V / iodide propidium were measured in response to PLCγ2 or PKC inhibitor treatment. Consistently, an increase in caspase-3 activity and significant apoptotic cell death were detected upon PLCγ2 or PKC inhibitor treatment in Nalm-6 and RS4;11 cells (Fig. 2f, g and Supplementary Fig. 3e–g).

In order to investigate the potential role of PLCγ2 in B-ALL resistance to GC, we generated a GC-resistant cell line by in vitro long-term exposure to dexamethasone (Dex). For this, Nalm-6 cells were cultured with an increasing concentration of Dex or an equivalent amount of DMSO (Dex solvent) for 44 days (Supplementary Fig. 4a). Then, cells were cultured in normal growth medium without Dex or DMSO for 14 days before any experiments. To determine Dex sensitivity, cells were seeded in 96-well plates for 24, 48, and 72 h in the presence of different concentrations of Dex (from 0 to 100 μM) and viability assay was performed. Dex-sensitive cells cultured in the presence of Dex became highly resistant to Dex (Nalm-6 R), while DMSO-treated cells were still sensitive (Nalm-6 S) (Fig. 2h and Supplementary Fig. 4b, c). Nalm-6 R cells were then treated with PLCγ2 inhibitors, i.e., U73122, edelfosine, 3-Nitrocoumarin, and manoalide, which all significantly decrease cell viability (Fig. 2i). A similar effect was observed in the presence of the PKC inhibitor GF109203X (Supplementary Fig. 4d). Together, these results indicate a role of PLC signaling in maintaining GC-resistant B-ALL cell survival.

### GC resistance is associated to higher PLCγ2 activation in B-ALL cells

Next, we investigated whether Dex resistance in Nalm-6 R cells was likely mediated by reduced expression of GR. Surprisingly, we found a higher expression of GR in resistant cells than in sensitive cells (Supplementary Fig. 4e). These data demonstrate that GR expression is not the only Dex sensitivity factor, and their relationship is not always correlated, in accordance with previous reports[43]. Since Nalm-6 R cells are sensitive to PLCγ2 inhibitors (Fig. 2i), we studied the activation of PLCγ2 using phospho flow cytometry assay. Interestingly, there was a strong PLCγ2 phosphorylation in resistant compared to sensitive cells (Supplementary Fig. 4f). PLCγ2 (Fig. 2j, k and Supplementary Fig. 4g, h) was not differentially expressed in sensitive and resistant B-ALL cells in publicly available transcriptomic dataset[36,43–45]. Also, analysis of paired data from publicly available datasets[46,47] showed no significant change in PLCγ2 expression between diagnosis and relapse in B-ALL (Supplementary Fig. 3h). In order to confirm the hyperactivation of PLC in resistant cells, we performed a direct fluorescence-based assay to detect PLC enzymatic activity. For this, we used Nalm-6 cells and Reh as a model for prednisone good responder (PGR) and for prednisone-poor responder patients, respectively, as classically described[48]. As expected, PLC enzyme activity was significantly higher in Reh cells than in Nalm-6 cells (Supplementary Fig. 4i, j). In addition, publicly available relapse data from B-ALL patient samples[46] revealed a signature of differentially expressed genes from diagnosis to relapse that is different between early (<36 months) and late (≥36 months) relapse. Based on this criterion, our unsupervised analysis[46] showed that PLCγ2 was significantly higher in early relapsed patients (≤34 months) compared to late relapsed (≥35 months) patients (Supplementary Fig. 4k). Collectively, these data suggest that Dex-resistant cells display a reliance on constitutively active PLC signaling.

### PLC inhibition dysregulates the transcriptional programs of cell cycle in Dex-resistant B-ALL cells

To define the function of PLC in GC-resistant B-ALL cells at the molecular level, we next performed transcriptome analysis in Ctr and U73122-treated-Nalm-6 R cells using RNA sequencing (RNA-seq). Of the 13,654 genes (Fig. 3a), 1085 genes were significantly differentially regulated in U73122-treated cells (±1.5-fold change in expression, adjusted $p$-value < 0.05), with 685 downregulated and 400 upregulated (Fig. 3b). Unbiased pathway enrichment analyses of differentially expressed genes (DEG) using Reactome, KEGG and Gene ontology (GO) analysis revealed the dysregulation of several signaling pathways in U73122-treated cells. Notably, 138 gene sets were significantly (p < 5%) under-enriched in Reactome analysis (Fig. 3c), 19 gene sets in KEGG analysis (Fig. 3e) and 508 gene sets in GO analysis (Fig. 3g), suggesting that PLC controls the transcriptional program of Nalm-6 R cells in different ways. Of these, the cell cycle-associated gene pathway was most significantly and consistently downregulated (Fig. 3c–h). Globally, transcriptional profiles of Ctr and U73122-treated-cell subsets were clearly distinct (Fig. 3i). Gene set enrichment analysis (GSEA) revealed upregulation of ER stress-induced intrinsic apoptotic signaling pathway-, phosphatase complex-associated genes; downregulation of B cell proliferation- and activation-associated pathways, such as cell cycle, Ca$^{2+}$ mediate signaling, regulation of cytosolic Ca$^{2+}$ ion concentration; and downregulation of PLC signaling-related pathways, such as PLC activating G-protein-coupled receptor pathway, PLC activity and ER Ca$^{2+}$ ion transport in the U73122-treated cells compared to the Ctr cells (Fig. 3j, k). The DEG in the transcription profiles also had coordinately altered cell viability, especially cell cycle-associated

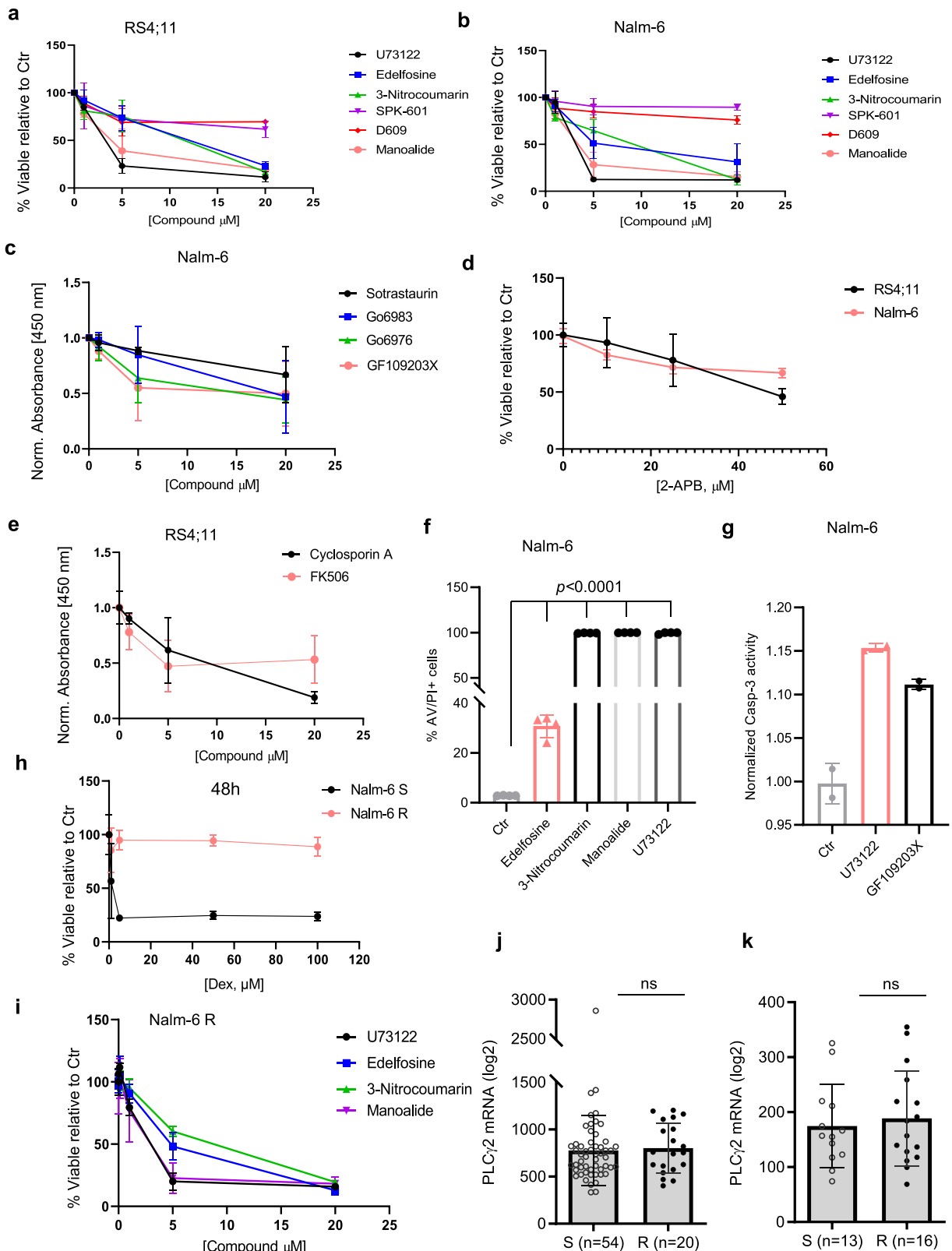

genes, such as *CDC6*, *CDK1*, *E2F1* and ER stress-associated genes, such as *XBP1*, *CHOP* (*Ddit3*), *PMAIP1*, *SIRT1*, and *SERINC3* (Fig. 3l). In order to understand the underlying regulatory mechanisms of this fundamental process, we also investigated how PLC inhibition disturbs the cell cycle process in Dex-sensitive B-ALL cells. Cells were treated with different PLC inhibitors, that all promoted cell cycle arrest at the subG1 phase while negatively regulated the G2/M phase (Supplementary

Fig. 5 and Supplementary Fig. 6a–f), in accordance with our GSEA analysis showing strong downregulation of genes involved in sister chromatid segregation mitotic at the G1/S transition and E2F-mediated G2/M phase cell cycle gene expression (Supplementary Fig. 6g). Collectively, these findings demonstrate that PLC promotes survival signaling in B-ALL cells by regulating cell cycle progression and division and beyond that, a complex program of gene expression.

**Fig. 2 | Inhibition of the PLC pathway triggers cytotoxicity in ALL cell lines and Dex-resistant cells. a, b** Cell viability relative to the Ctr of RS4;11 (**a**) and Nalm-6 cells (**b**) after 48 h exposure to increasing concentrations of different PLC inhibitors (*n* = 3 independent experiments). **c** Viability normalized to the Ctr of Nalm-6 cells after 48 h exposure to increasing concentrations of different PKC inhibitors (*n* = 3 independent experiments). **d** Viability relative to the Ctr of Nalm-6 cells and RS4;11 cells after 48 h exposure to increasing concentrations of 1,4,5-trisphosphate (IP$_3$)-induced Ca$^{2+}$ release inhibitor (*n* = 4 independent experiments). **e** Viability normalized to the Ctr of RS4;11 cells after 48 h exposure to increasing concentrations of calcineurin inhibitors (*n* = 4 independent experiments). **f** Frequency of annexin V/PI-positive Nalm-6 cells determined by flow cytometry after 24 h of exposure to PLC inhibitors (*n* = 4 independent experiments). **g** Caspase-3 activity in Nalm-6 cells determined by using Ac-DEVD-AFC as substrate after 24 h of exposure to PLC and PKC inhibitors. Data are mean ± SEM *n* = 2 independent experiments, performed in 7 technical replicates. **h** Viability relative to the Ctr of Dex-resistant Nalm-6 cells (referred to as Nalm-6 R) and Dex-sensitive Nalm-6 cells (referred to as Nalm-6 S), exposed for 48 h with increasing concentrations of Dex (*n* = 4 independent experiments). **i** Viability relative to the Ctr of Nalm-6 R exposed for 48 h to increasing concentrations of PLC inhibitors (*n* = 3 independent experiments). **j, k** RNA-seq analysis of PLCγ2 expression in glucocorticoid-sensitive (S) or -resistant (R) primary B-ALL. Data are presented as mean ± SD and were extracted from publicly available transcriptomic dataset[44,45]. All data were analyzed using a standard Student's *t*-test (two-sided). Unless otherwise indicated, data are presented as mean ± SEM. Source data are provided as a Source Data file.

## Dex exposure triggers Ca$^{2+}$ signaling through the PLCγ2 pathway in B-ALL cells

Given the importance of PLCγ2 for B cell function[10,15,16,18,19] and intracellular Ca$^{2+}$ signals for B-ALL cell resistance to Dex[8], we investigated the role of PLCγ2 in Dex-mediated Ca$^{2+}$ signaling in B-ALL. To optically measure cytosolic Ca$^{2+}$ shifts, we used single-cell Ca$^{2+}$ imaging by fluorescence microscopy following stimulation of B-ALL cells with Dex (Fig. 4a). Dex exposure resulted in increased cytosolic Ca$^{2+}$ in B-ALL cells (Fig. 4a–c and Supplementary Fig. 7a, b). Additionally, these observations were also validated using Fluo-4 AM probe (Supplementary Fig. 7c). Consistent with our previous reports[8], we found that robust and sustained cytosolic Ca$^{2+}$ signals induced by Dex were significantly higher in 4 mM extracellular Ca$^{2+}$ as compared with Ca$^{2+}$-free buffer (0 mM extracellular Ca$^{2+}$, Fig. 4b, c and Supplementary Fig. 7a, b). Since this pathway of Ca$^{2+}$ entry into B-ALL cells bears the pharmacological characteristics of store-operated Ca$^{2+}$ entry (SOCE), we measured SOCE by stimulating cells with Dex in Ca$^{2+}$-free buffer, which induces activation of SOC channels following ER Ca$^{2+}$ depletion, and SOCE was then evaluated by the addition of 4 mM extracellular Ca$^{2+}$. Indeed, Dex significantly induced ER Ca$^{2+}$ release and SOCE (Supplementary Fig. 7d–g), suggesting that Dex probably regulates SOCE via Ca$^{2+}$ release from the ER via IP$_3$ receptor channels following PLC activation. To test this hypothesis, we treated B-ALL cells with Dex for 5 min, the time required to induce a sustained Ca$^{2+}$ increase, and measured PLCγ2 activation by phospho flow cytometry. Dex stimulation triggered a strong upregulation of phospho-PLCγ2 in B-ALL cell lines (Fig. 4d, e and Supplementary Fig. 8a, b). Moreover, to validate the ability of Dex to activate PLC in B-ALL cells, we performed a direct fluorescence-based assay for detecting protease activity after 24 h of Dex exposure and showed that Dex increased PLC activity in B-ALL cells, which was decreased in the presence of PLC inhibitor (Fig. 4f, g). To ensure that PLC activation by Dex was functional, we measured Dex-mediated Ca$^{2+}$ in the presence of different PLC inhibitors. Dex-mediated Ca$^{2+}$ signaling was prevented by PLC inhibitors, which was evident from the decrease in Ca$^{2+}$ released from ER stores and the overall Ca$^{2+}$ entry from stores including external and internal (Fig. 4h–m and Supplementary Fig. 8c–e). Most importantly, the data from the genetic silencing of PLCγ2 as well as PLCγ1 were consistent with the pharmacological inhibitor features of Dex-induced B-ALL cell death (Supplementary Fig. 9a–d). Collectively, these results demonstrate that Dex-induced Ca$^{2+}$ response in B-ALL cells is mainly mediated by PLC.

## PLCγ2 signaling mediates Dex resistance in B-ALL cells

Increased cytosolic Ca$^{2+}$ through PLCγ2 in both normal and pathological B cells has been shown to induce cell survival and growth[10,15,18]. Since Dex exposure activates PLCγ2 signaling pathway, we aimed to determine whether PLCγ2 inhibition could enhance Dex sensitivity and reverse Dex resistance in B-ALL cells. For this, we treated B-ALL cell lines with pharmacological or genetic inactivation of PLCγ2 as well as PLCγ1 in combination with Dex. Cell death was strongly increased by combined treatments (Fig. 5a, b; Supplementary Figs. 9e, f 10a). The

ability of PLC inhibition to increase GC sensitivity was confirmed by LIVE/DEAD® (Fig. 5c, d) and annexin V/PI (Fig. 5e, f) staining. Moreover, these observations were validated in primary blood and bone marrow (BM) samples from a cohort of 44 ALL patients with highly variable in vitro Dex sensitivity (Supplementary Fig. 10b). By defining Dex sensitivity (≥20% cell death) and Dex resistance (<20% cell death) following Dex exposure, we found that 80% and 72% primary diagnostic blood and BM samples were intrinsically Dex resistant when cultured in vitro in the presence of 100 nM Dex, respectively (Supplementary Fig. 10b). Importantly, inhibition of PLC signaling with U73122 was sufficient to overcome Dex resistance in 62% and 52% of these Dex-resistant blood and BM samples, respectively (Fig. 5g). Consistent with the decrease in cell viability, the combination of Dex and the PLC inhibitor showed a decrease in cell proliferation, as evidenced by decreased CTV dilution (Fig. 5j, k).

To further investigate the mechanistic basis of Dex resistance mediated by PLC signaling, we next assessed the role of PLC downstream signaling pathways. PLC activation leads to the production of IP$_3$ and DAG, which subsequently and respectively trigger the release of ER Ca$^{2+}$ via the IP$_3$R channel and PKC activation[40]. We examined the roles of these two downstream cascades, the DAG/PKC-dependent and IP$_3$/Ca$^{2+}$-dependent pathways. For this, we first used the IP$_3$R channel inhibitor 2-APB[42], which inhibits ER Ca$^{2+}$ output. As expected, dead cells were greatly increased by co-treatment with 2-APB and Dex (Supplementary Fig. 10c, d). To further establish a direct connection between Dex and PLC signaling, we analyzed PKC activity, which is activated by PLC through DAG by using an enzyme-linked immunosorbent (ELISA) assay. Dex stimulation resulted in strong upregulation of PKC activity, which was reduced by PLC inhibitor U73122 (Supplementary Fig. 10e), suggesting that PKC is indeed activated downstream of Dex-induced PLC signaling. Moreover, different PKC inhibitors significantly increased Dex-mediated B-ALL cell death (Supplementary Fig. 10f, g). The ability of PKC inhibition to increase Dex sensitivity was confirmed by LIVE/DEAD® staining (Supplementary Fig. 10h, i). Consistent with the results in B-ALL cell lines, the PKC inhibitor GF109203X restored Dex sensitivity in Dex-resistant primary blood and BM samples (Supplementary Fig. 10j). After having established that the cell cycle was regulated by the PLC pathway, we determined whether this increase in cell death was in part caused by cell cycle arrest. Treatment of B-ALL cells with U73122, GF106203X, and 2-APB resulted in increased Dex-induced cell death, as manifested by an increase in subG1 apoptotic cell population (Fig. 5h, i and Supplementary Fig. 11a, b). These results strongly suggest that the PLC pathway plays a critical role in inducing GC resistance, and its activity can be targeted to sensitize B-ALL cells to GC treatment.

We next asked whether PLC inhibition could affect GR transcriptional program and expression. GSEA analysis of our RNA-seq revealed an enrichment of the GR pathway and the cellular response to Dex stimulus genes in U73122-treated cells (Fig. 5l and Supplementary Fig. 11c–e) and an increase in GR expression after treatment with manoalide (Supplementary Fig. 11f, g), further indicating that the sensitivity of B-ALL cells to GC is dependent on PLC function.

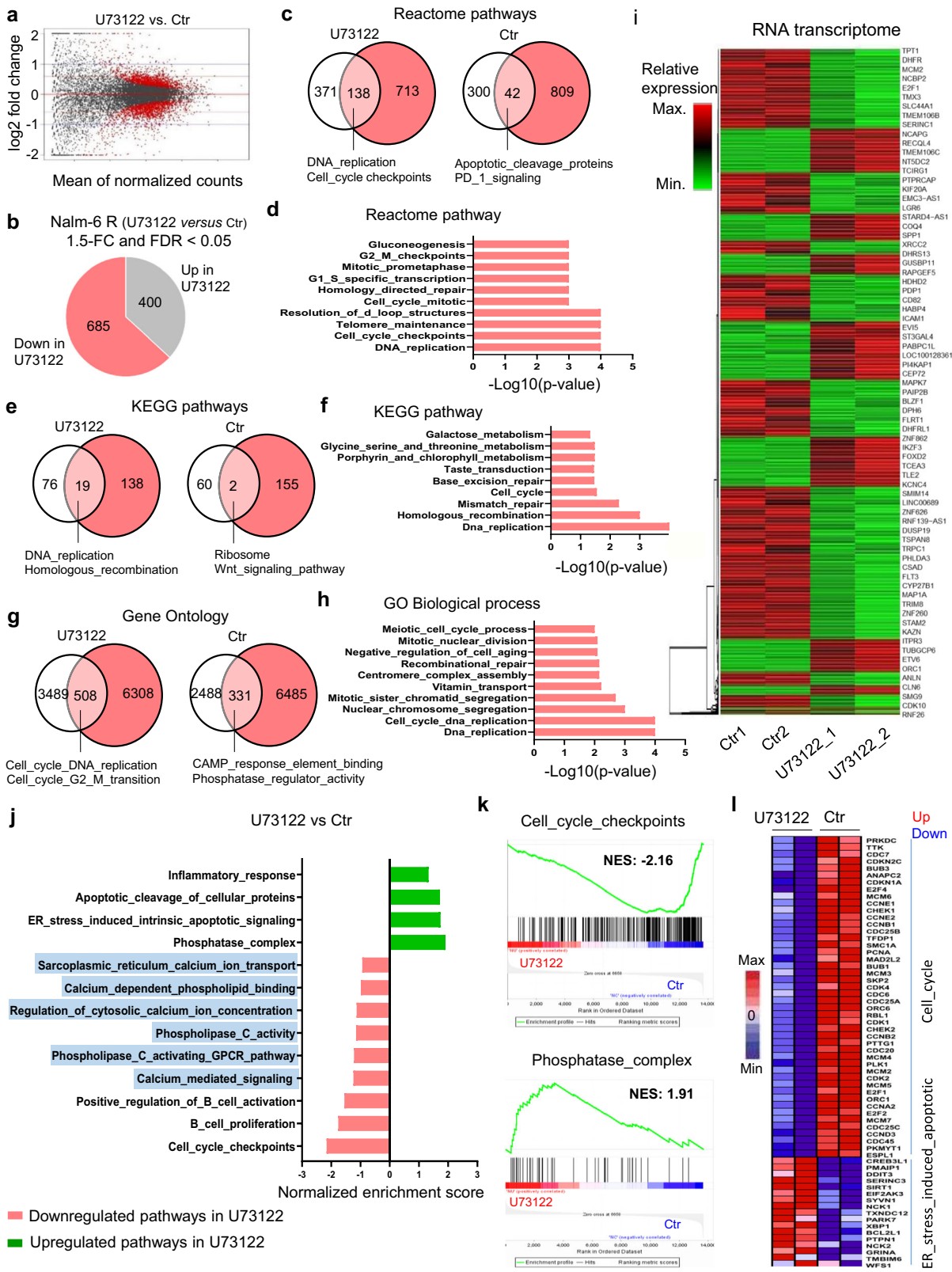

We next focused our attention on the calcineurin/NFAT signaling, another downstream pathway controlled by PLC through IP$_3$-mediated Ca$^{2+}$ signaling activation[25]. We investigated whether this pathway could affect GC sensitivity in B-ALL cells. We found that calcineurin inhibitors (cyclosporin A and tacrolimus, FK506) also significantly increased Dex sensitivity in B-ALL cell lines (Supplementary Fig. 12). To assess whether PLCγ2 inhibition could reverse Dex resistance, we used our strongly Dex-resistant Nalm-6 R cells (Fig. 2h and Supplementary Fig. 4b, c). PLC, PKC, and calcineurin inhibitors were sufficient to overcome Dex resistance (Supplementary Fig. 13a–g). Consistent with previous reports[45], inhibiting mTOR by sirolimus/rapamycin or Akt by MK-2206 (Supplementary Fig. 13h), two pathways interconnected with PLCγ2/Ca$^{2+}$/PKC signaling and implicated in B cell development and cancer[10,49], also reversed Dex resistance. Together, these data provide

**Fig. 3 | Pharmacological inhibition of PLC alters gene expression in Dex-resistant Nalm-6 (Nalm-6 R).** **a** MA plot showing the relationship between log2-fold changes of RNA reads in U73122 versus Ctr Nalm-6 R cells plotted against mean expression values. Red dots represent genes with an adjusted $p$-value < 0.05 using DESeq2's differential expression test. **b** Differentially expressed genes (DEG) in Nalm-6 R cells after 16-h exposure to 1 μM U73122 determined by RNA-seq analysis (false discovery rate [FDR] adjusted $p < 0.05$, and fold change (log2 scale) ≥1.5 or ≤ −1.5) using DESeq2 test. **c, e, g** Summary of Reactome (**c**), KEGG (**e**), and GO (**g**) pathway analysis of Ctr and U73122-treated cells. Venn diagram shows gene sets that are downregulated, non-significant (white), significant with p-value < 0.05 (pink), and unregulated (red) as determined by gene set enrichment analysis (GSEA) Kolmogorov–Smirnov test. Examples of unique pathways are indicated for each group. **d, f, h** Functional clustering of downregulated genes in U73122-exposed-Nalm-6 R cells ($n = 2$ biological replicates) with FDR adjusted $p$-value < 0.05 for KEGG (**f**), <0.001 for Reactome (**d**), <0.03 for GO Biological process (**h**) using GSEA (Kolmogorov–Smirnov) test. Some top pathways are indicated. **i** Hierarchical clustering after RNA-seq analysis showing DEG calculated by DESeq2 test in Nalm-6 R cells after U73122 exposure ($n = 2$ biological replicates). **j** Normalized enrichment scores of significantly up- or downregulated gene sets in Nalm-6 R cells after U73122 exposure ($n = 2$ biological replicates). **k** Representative GSEA enrichment plot showing upregulation of phosphatase_complex-related genes and downregulation of cell cycle checkpoint-related genes in Nalm-6 R cells after U73122 exposure ($n = 2$ biological replicates). **l** Heatmap of ER_stress_induced_apoptotic and cell cycle DEG in Nalm-6 R cells after U73122 exposure ($n = 2$ biological replicates). Source data are provided as a Source Data file.

evidence that GC paradoxically induce their own resistance by activating the pro-survival signaling of the PLC/Ca$^{2+}$/PKC pathway.

## CXCR4 is highly expressed in B-ALL cells and positively correlated with PLCγ2

We hypothesized that the activation of PLCγ2 signaling by Dex was mediated by cell surface receptor signaling. A whole-genome correlation network analysis of PLCγ2 using the R2 database identified 281 genes whose transcript expression strongly correlated with those of PLCγ2 in MILE dataset[35] (Fig. 6a, b, Supplementary Fig. 14a, Pearson's $r > 0.5$, $p < 1.8E-125$). In an attempt to identify genes correlated with PLC but also involved in Ca$^{2+}$ signaling, we compared these genes co-expressed with PLC to the public Ca$^{2+}$ signature genes from the Gene Ontology Biological Process (GOBP) Gene Set (Supplementary Fig. 14b). We found that six of these genes were involved in Ca$^{2+}$-mediated intracellular signal transduction (Fig. 6a, b). With the exception of the *PPP3CC* (Protein Phosphatase 3 Catalytic Subunit Gamma) gene, the other five are all genes encoding cell surface proteins in humans. Of the six identified targets, CXCR4 mRNA was the most expressed in the 12 B-ALL cell lines available in the CCLE dataset[36] (Fig. 6c), whereas *CD22* was moderately expressed, and the expression of *SLC9A1* (Solute Carrier family 9A1), *CCR7* (C-C Motif Chemokine Receptor 7), *PPP3CC* and *CXCR5* (C-X-C Motif Chemokine Receptor 5) was very low or undetectable in these cell lines.

In addition to the positive correlation between *PLCγ2* and *CXCR4* observed in MILE leukemia cohort[35] (Supplementary Fig. 14c, Pearson's $r = 0.529$, $p = 1.11E-139$), these two genes were also positively correlated in the DLBCL dataset[39] (Supplementary Fig. 14d, Pearson's $r = 0.51$, $p = 0.00027$). We then extended our comparison of the *CXCR4* gene in human B-ALL to other cancer types[36]. mRNA levels of CXCR4 were more highly expressed in B-ALL than in most other human cancers (ranked 3 of 40) (Fig. 6d and Supplementary Fig. 15a). This expression was associated with hypomethylation in human B-ALL compared to most other cancers (ranked 37 of 38) (Supplementary Fig. 15b, c), an epigenetic modification observed in B-ALL[50] well described as a tumor stimulator that promotes mitotic cell division[51]. Analysis of publicly available data revealed that CXCR4 mRNA was highly expressed in primary samples of B-ALL and T-ALL compared to normal samples[52] (Supplementary Fig. 15d). This increase was also observed in the GC-resistant B-ALL primary samples compared to the sensitive ones[45] (Supplementary Fig. 15e). We next performed a meta-analysis using R2 database survival analysis to further assess the role of CXCR4 in clinical outcomes. The analysis revealed that cancers with higher CXCR4 expression had significantly worse overall survival (Supplementary Fig. 15f, i). Collectively, these data establish the importance of the *CXCR4* pathway in B-ALL.

## CXCR4 mediates Dex-induced PLCγ2/Ca$^{2+}$ axis activation in B-ALL cells

We next assessed whether the correlation between PLCγ2 and CXCR4 had functional consequences in B-ALL cell signaling and whether it was critical for Dex sensitivity. We first checked the expression of CXCR4

on our cell models. CXCR4 is highly expressed at the surface of B-ALL cell lines and downregulated upon CXCR4 blockade using small molecule antagonists (Supplementary Fig. 16a–d). PLCγ2 has previously been shown to play a role in CXCR4-mediated B cell migration[16]. CXCR4 antagonist alone had no effect on Ca$^{2+}$ signaling (Supplementary Fig. 16e). The natural ligand of CXCR4 (SDF-1α) exposure resulted in increased cytosolic Ca$^{2+}$ in B-ALL cells, which was abrogated in the presence of CXCR4 antagonists (Supplementary Fig. 16f–m). These results indicate that CXCR4 is functionally active in B-ALL cells. Next, we investigated the role of CXCR4 in the regulation of Dex-induced intracellular Ca$^{2+}$ signals. Interestingly, we found that the robust and sustained cytosolic Ca$^{2+}$ signals induced by Dex were significantly curtailed in the presence of CXCR4 antagonists (Fig. 6e, f and Supplementary Fig. 16n–s). To validate the role of CXCR4 in Ca$^{2+}$ signaling of Dex, in a first approach, we negatively regulated the expression of CXCR4 using small interfering RNA (siRNA) in Nalm-6 cells (Fig. 6g–i). Correlatively, Dex-induced Ca$^{2+}$ signaling upon Dex exposure was decreased in Nalm-6 transfected with siRNA CXCR4 (siCXCR4) as compared to siCtr (Fig. 6h, i). As a second approach, we mutated endogenous CXCR4 in two B-ALL cell lines cells using CRISPR/Cas9 with Green fluorescent protein (GFP), a marker of cell transfection with the CRISPR-Cas9 plasmid targeting CXCR4 (CRISPR-Cxcr4) (Supplementary Fig. 16t). B-ALL cells were then sorted for GFP positivity. Flow cytometry assays indicated a substantial decrease in CXCR4 surface expression in CRISPR-Cxcr4 cells compared with CRISPR-Ctr cells (Supplementary Fig. 16u, v). Importantly, Ca$^{2+}$ signaling induced by SDF-1 was completely suppressed in the CRISPR-Cxcr4 cells, indicating the non-functionality of the CXCR4 receptor in the transfected cells (Fig. 6j, k and Supplementary Fig. 16w, x). Subsequently, we measured Dex-induced Ca$^{2+}$ signaling in these transfected cells and found that Dex did not induce increases in the cytosolic Ca$^{2+}$ in CRISPR-Cxcr4 cells, demonstrating that CXCR4 controls Dex-induced Ca$^{2+}$ signaling in B-ALL cells (Fig. 6l, m and Supplementary Fig. 16y, z). Since CXCR4 is coupled to G-protein-mediated signaling pathway, which regulates Ca$^{2+}$ signaling primarily via IP$_3$ production and Ca$^{2+}$ release from the ER via IP$_3$R channels, we evaluated this by stimulating cells with Dex in Ca$^{2+}$-free buffer in the presence of CXCR4 antagonists. We observed decreased ER Ca$^{2+}$ release in CXCR4 antagonist-treated cells compared to Ctr in response to Dex stimulation (Supplementary Fig. 17a–d). We also measured SOCE as described above. As expected, Dex-mediated SOCE was significantly decreased in the presence of CXCR4 antagonists (Supplementary Fig. 17e–h). Since SOCE and IP$_3$-mediated ER Ca$^{2+}$ release are triggered by PLC activation, we treated B-ALL cells with Dex for 5 min, the time required to induce a sustained Ca$^{2+}$ increase, and measured PLCγ2 activation by phospho flow cytometry in the presence of CXCR4 antagonists, CRISPR-Cxcr4 or siCXCR4. Dex triggered a strong upregulation of phospho-PLCγ2 in B-ALL cell lines, which was impaired in CRISPR-Cxcr4 and siCXCR4 cells as well as after exposure to CXCR4 antagonists (Fig. 6n, o and Supplementary Fig. 17i–l). Total PLCγ2 expression is

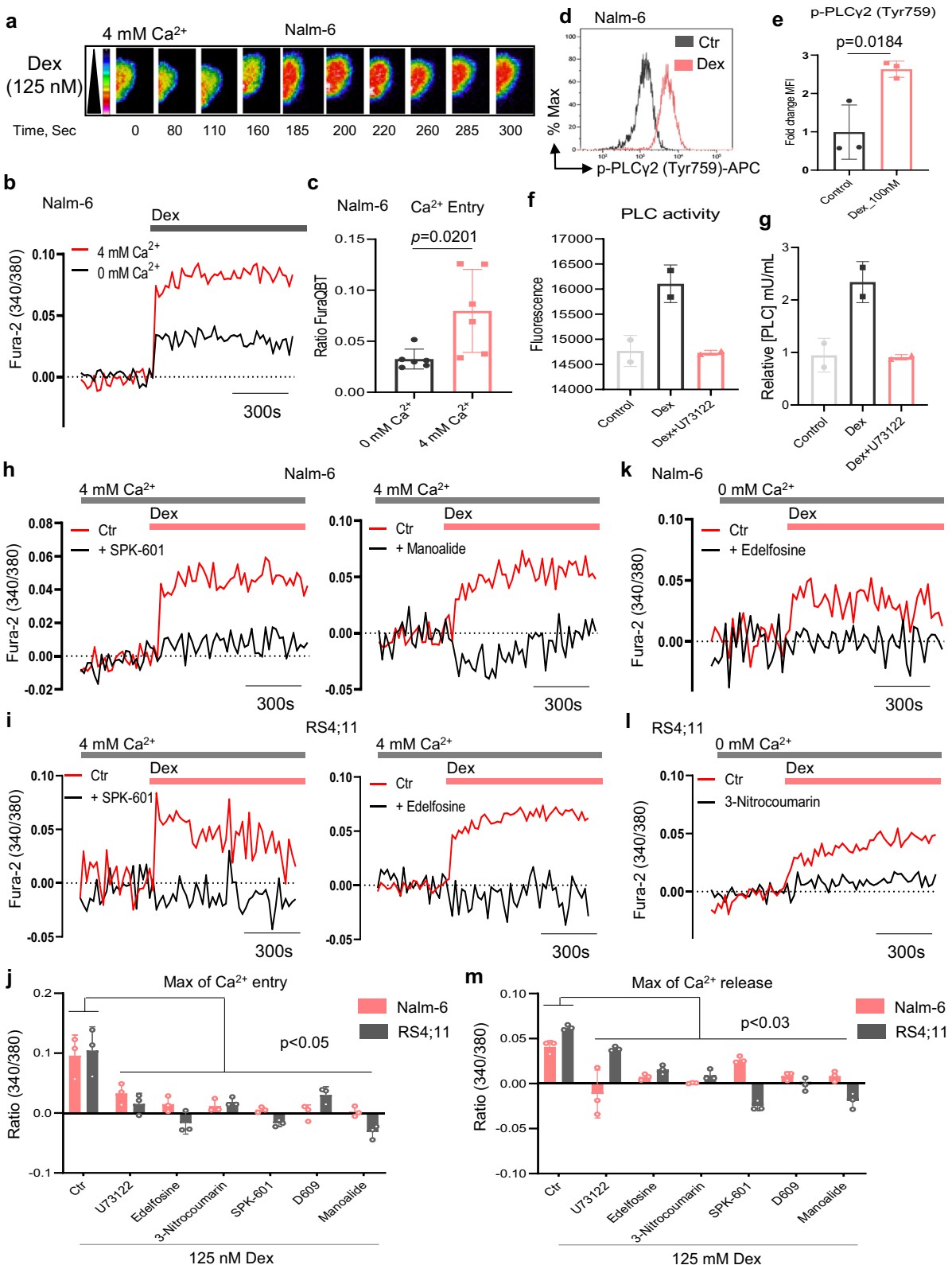

not affected by CXCR4 antagonists or CRISPR-Cxcr4 (Supplementary Fig. 17m–p). Collectively, these results demonstrate that Dex-induced Ca²⁺ response in B-ALL cells is mainly mediated by PLCγ2 through CXCR4.

Because previous studies established that the CXCR4 receptor is rapidly internalized upon activation[53,54], we determined whether Dex-induced CXCR4 internalization in B-ALL cells. As a first approach, we

treated B-ALL cells with Dex for 5 min and measured CXCR4 surface expression by flow cytometry. As shown in Fig. 6p–s and Supplementary Fig. 17q, r, Dex stimulation triggered a strong downregulation of CXCR4 surface expression in four B-ALL cell lines, i.e., (Nalm-6, Reh, RS4;11 and HAL-01), consistently with previous reports demonstrating that CXCR4 internalization occurs 2–5 min after activation[53,55]. The downregulation of CXCR4 induced by Dex may be due in part to its

**Fig. 4 | Dexamethasone (Dex) triggers PLC-mediated Ca²⁺ signaling in ALL cells.** **a** Colored time-lapse images of Nalm-6 cells show the changes in cytosolic Ca²⁺ evoked by Dex in Ca²⁺-containing buffer. **b** Cytosolic Ca²⁺ measurements in Nalm-6 cells stimulated with 125 nM Dex in nominally Ca²⁺-free buffer (0 mM Ca²⁺) and Ca²⁺-containing buffer (4 mM Ca²⁺). **c** Quantification of maximal Ca²⁺ entry from (**b**). Data shown are the mean ± SEM ($n = 6$ independent experiments). **d** Flow cytometry plots of the active form of PLCγ2 (phospho-PLCγ2 (Tyr759)) stimulated with Dex or Ctr for 5 min. **e** Quantification from (**d**) normalized to Ctr. Data are mean ± SEM ($n = 3$ independent experiments). **f, g** Quantification of PLC activity (**f**) and intra-cellular concentration of PLC (mU/mL) (**g**) in Nalm-6 cells stimulated with 100 nM Dex or Ctr for 24 h with or without 1 μM U73122 and normalized to Ctr in (**g**). Data are mean ± SEM from $n = 2$ independent experiments, performed in technical octuplicates. **h, i** Cytosolic Ca²⁺ measurements in Nalm-6 (**h**) and RS4;11 (**i**) cells stimulated with 125 nM Dex in Ca²⁺-containing buffer (4 mM Ca²⁺). Cells were pre-incubated with SPK-601, and either Manoalide (**h**) or Edelfosine (**i**) during Fura-2 loading before stimulation. All inhibitors are used at 10 μM. **j** Maximal cytosolic Ca²⁺ entry from (**h, i**). Data are mean ± SEM ($n = 3$ independent experiments). Exact P-values from two-way ANOVA with Sidak's multiple comparisons test were as fol-lows: Nalm-6: Ctr vs. U73122, $p = 0.0491$; Ctr vs. Edelfosine, $p = 0.0024$; Ctr vs. 3-Nitrocoumarin, $p = 0.0014$; Ctr vs. SPK-601, $p = 0.0005$; Ctr vs. D609, $p = 0.0002$; Ctr vs. Manoalide, $p = 0.0003$. RS4;11: Ctr vs. U73122, $p < 0.0001$; Ctr vs. Edelfosine, $p < 0.0001$; Ctr vs. 3-Nitrocoumarin, $p < 0.0001$; Ctr vs. SPK-601, $p < 0.0001$; Ctr vs. D609, $p = 0.0002$; Ctr vs. Manoalide, $p < 0.0001$. **k, l** Traces of ER Ca²⁺ release in Nalm-6 (**k**) and RS4;11 (**l**) cells stimulated with 125 nM Dex in Ca²⁺-free buffer (0 mM Ca²⁺). Cells were preincubated with either 10 μM Edelfosine (**k**) or 10 μM 3-Nitrocoumarin (**l**) during Fura-2 loading before stimulation. **m** Maximal ER Ca²⁺ release from (**k, l**). Data are mean ± SEM ($n = 3$ independent experiments). Exact P-values from two-way ANOVA with Sidak's multiple comparisons test were as fol-lows: Nalm-6: Ctr vs. U73122, $p < 0.0001$; Ctr vs. Edelfosine, $p = 0.0004$; Ctr vs. 3-Nitrocoumarin, $p = <0.0001$; Ctr vs. SPK-601, $p = 0.0092$; Ctr vs. D609, $p = 0.0007$; Ctr vs. Manoalide, $p = 0.0006$. RS4;11: Ctr vs. U73122, $p = 0.0267$; Ctr vs. Edelfosine, $p < 0.0001$; Ctr vs. 3-Nitrocoumarin, $p < 0.0001$; Ctr vs. SPK-601, $p < 0.0001$; Ctr vs. D609, $p < 0.0001$; Ctr vs. Manoalide, $p < 0.0001$. Data (in **c, e–g**) was analyzed by two-tailed, unpaired Student's t-test. Source data are provided as a Source Data file.

internalization rather than its inhibition because Dex induces Ca²⁺ signaling, which is blocked by the inhibition of CXCR4. In addition, based on the idea that cold impedes the internalization of membrane receptors[54], we stimulated B-ALL cells with SDF-1α or Dex and imme-diately transferred them on ice (at 4 °C, $T = 0$) or incubated at 37 °C for 1 or 2 h ($T = 1, 2 h$). As shown in Fig. 6t, CXCR4 cell surface expression on B-ALL cells was downregulated after 1 h or 2 h of exposure to Dex or SDF-1α at 37 °C. As shown in Supplementary Fig. 17s–u, and Fig. 6u, Dex-mediated downregulation of CXCR4 expression was also observed over time (up to 20–24 h after stimulation), though the effect was reduced, suggesting that the receptor recycles poorly after sti-mulation, as demonstrated for SDF-1[56]. We next investigated whether Dex-mediated CXCR4 internalization was a driver or an outcome of its activation, by determining if Dex could induce Ca²⁺ signaling when CXCR4 internalization was prevented (i.e., at +4 °C rather than +37 °C). We found that in the absence of CXCR4 internalization, Ca²⁺ signaling was indeed triggered by Dex, confirming that internalization of CXCR4 was just a secondary effect of its activation (Supplementary Fig. 18a–e). Furthermore, the GC receptor antagonist (RU486) had no effect on Dex-mediated Ca²⁺ signaling (Supplementary Fig. 18f). In a second approach, we investigated whether Dex could interfere with the ability of SDF-1 to stimulate Ca²⁺ signaling in B-ALL cells. Interest-ingly, when SDF-1 was added after Dex, i.e., at the time of maximum Ca²⁺ peak (Fig. 6v, w and Supplementary Fig. 19a–d), no additive effect on Ca²⁺ entry was observed, suggesting that Dex is an effective competitor of SDF-1 for CXCR4 activation of Ca²⁺ signaling. Other-wise, when Dex was added after SDF-1 (Supplementary Fig. 19e, f), a slight but non-significant additive effect on Ca²⁺ entry was observed. These results suggest that the unresponsiveness of SDF-1 can be mainly attributed to CXCR4 receptor internalization by Dex. To confirm this, we used another molecule thapsigargin (TG), capable of stimulating Ca²⁺ (Supplementary Fig. 19g, h) by blocking sarcoplasmic-endoplasmic Ca²⁺-ATPase, therefore a different path-way to CXCR4, as shown in Supplementary Fig. 19i, j, where the CXCR4 antagonist AMD3100 had no effect on ER Ca²⁺ release and SOCE induced by TG. As suspected, when TG was added after Dex (Supplementary Fig. 19k–r), or when Dex was added after TG (Sup-plementary Fig. 19s–v), the effect on Ca²⁺ entry was additive, sug-gesting that Dex acted to mobilize Ca²⁺ via a CXCR4-specific and dependent mechanism. Since CXCR4 is localized in lipid rafts[57] and localization in lipid rafts can impact Ca²⁺ signaling by influencing G-protein and PLC activation, we disrupted lipid rafts by methyl-β-cyclodextrin (M-βCD) and showed a decrease in Dex-induced Ca²⁺ signaling in B-ALL cell lines and patient samples (Supplementary Fig. 19w–z). Collectively, these results demonstrate that Dex stimu-lates CXCR4 signaling in B-ALL cells and thereby regulates PLCγ2 activation and signaling.

## CXCR4 inhibition enhances Dex sensitivity in B-ALL cell lines in vitro and in vivo

To assess whether Dex-mediated CXCR4 stimulation could have functional consequences on B-ALL cell survival, we first assessed the oncogenic activity of CXCR4 in human cancers using the Project Achilles database[58], which is a genome-wide shRNA library to identify genes that affect cancer cell survival and proliferation. In this analysis, we found that B-ALL showed a negative shRNA score in response to CXCR4 shRNA knockdown but which was also much lower than in most other human cancers (ranked 29 of 31) (Fig. 7a and Supple-mentary Fig. 20a), indicating that CXCR4 pathway is necessary for B-ALL cell survival in accordance with previous report[33]. Furthermore, in line with previous findings[34], CXCR4 activation by SDF-1 increased the viability of B-ALL cells (Supplementary Fig. 20b). We next investi-gated whether SDF-1-mediated B-ALL cell viability modulated Dex sensitivity. This analysis revealed a reduction in Dex sensitivity in the presence of SDF-1 (Supplementary Fig. 20c, d). The dose-response graph of SDF-1 indicated that a concentration as low as 5 ng/mL ren-dered HAL-01 B-ALL cells completely resistant to Dex (Supplementary Fig. 20e). Interestingly, CXCR4 antagonists were sufficient to restore Dex sensitivity in the presence of SDF-1 (Supplementary Fig. 20f), indicating the functionality of its receptor, CXCR4. Aiming to validate the potential role of CXCR4 signaling on the sensitivity of B-ALL cells to Dex, we next treated B-ALL cell lines with different CXCR4 antagonists, i.e., AMD3100, AMD3465, WZ811 or MSK122, in combi-nation with Dex (Supplementary Fig. 20g, h). B-ALL cells were treated with CXCR4 antagonists at doses that did not affect cell growth. In this analysis, B-ALL cell death was strongly increased by combined treat-ments (Fig. 7b and Supplementary Fig. 20i, j). The ability of CXCR4 signaling inhibition to increase GC sensitivity was confirmed by annexin V/PI staining (Fig. 7c–f) in CRISPR-Cxcr4 cells as well as in siCxcr4 Nalm-6 cells (Supplementary Fig. 20k). Moreover, inhibition of CXCR4 signaling with AMD3100 was sufficient to overcome Dex resistance in 37% and 48% of Dex-resistant blood and BM samples, respectively (Fig. 7g). It should be noted that these CXCR4 antagonists had no effect on GR (Supplementary Fig. 21a, b), and that GR expression was not altered in CRISPR-Cxcr4 cells (Supplementary Fig. 21c, d). To apprehend the mechanisms leading to the improve-ment of Dex sensitivity mediated by CXCR4 signaling inhibition, we determined whether this increase in cell death was associated with cell cycle arrest. Treatment of B-ALL cells with AMD3100 resulted in increased Dex-induced cell death, as manifested by a significant amount of subG1 apoptotic cell population (Fig. 7h and Supplemen-tary Fig. 21e, f). We next investigated the effect of inhibiting CXCR4 and PLC in vivo. For this, Nalm-6-luciferase-GFP cells were injected in immunodeficient NSG mice. By using flow cytometry Ca²⁺ assessment in Nalm-6 GFP⁺ cells derived from the blood of mice at day 27 (Fig. 8a),

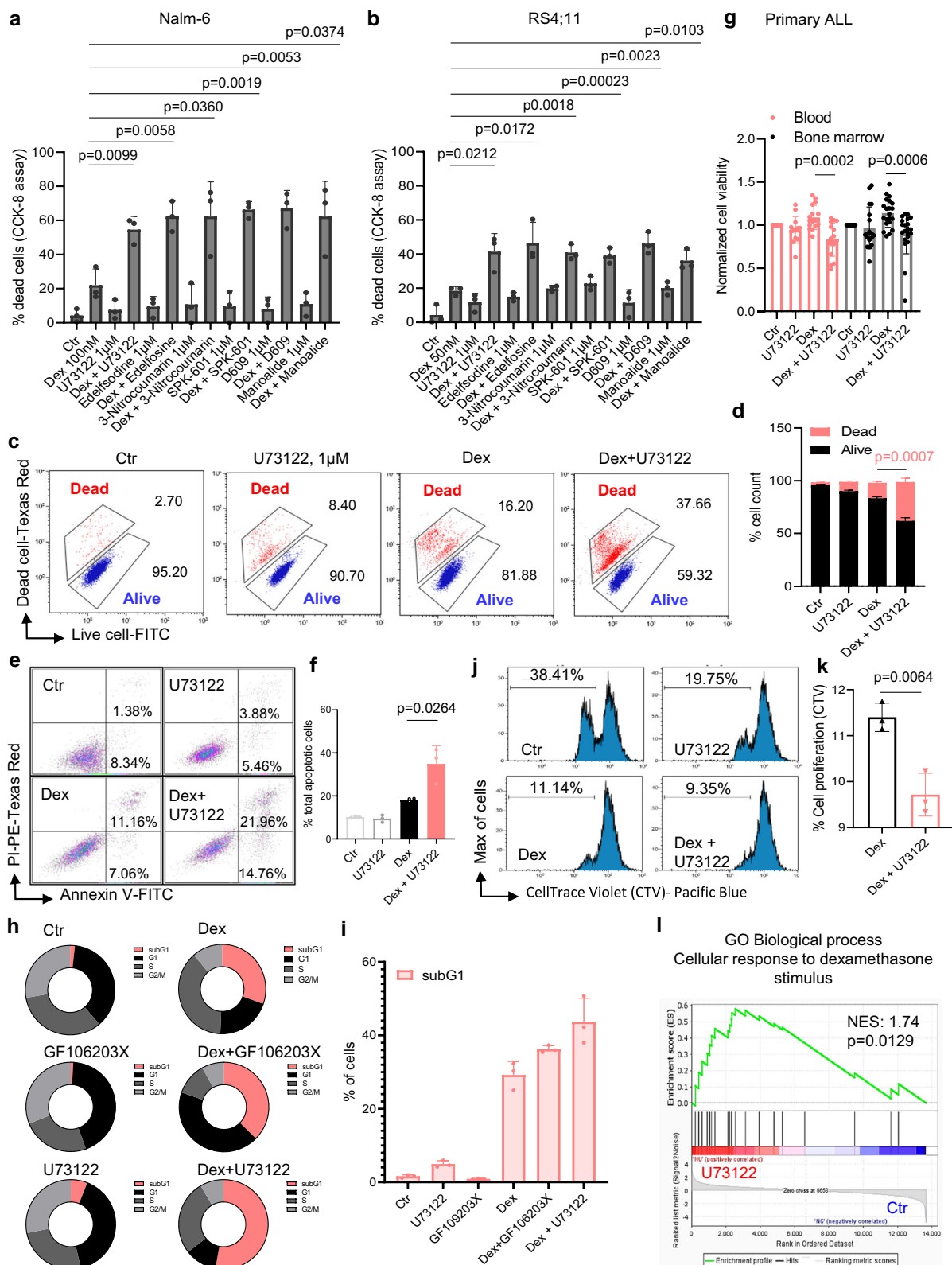

we observed that the increase in Ca²⁺ induced by Dex treatment was reduced in the presence of AMD (Fig. 8b). Consistently, Dex treatment prolonged mouse overall survival (Fig. 8c, d and Supplementary Fig. 21g). Importantly, AMD3465 further ameliorated survival indicating that blocking CXCR4 signaling in vivo augments sensitivity to Dex by regulating Ca²⁺ signaling. To further confirm the relevance of the proposed signaling axis and to improve the translatability of our work

in GC-resistant preclinical models, we used two in vivo approaches. First, we made our Nalm-6-LUC/GFP cells resistant to GC (method described in Supplementary Fig. 4a). Supplementary Fig. 21h shows the validation of the model in the NSG mouse model. We then tested this resistant line and showed similar results to the sensitive line (Fig. 8e–h). We also showed in this experiment that PLC inhibition with U73122 improved mouse survival in the presence of Dex (Fig. 8e–h).

**Fig. 5 | PLC Inhibition enhances Dex sensitivity in ALL cell lines and overcomes Dex resistance in Dex-resistant cells. a, b** Nalm-6 (**a**) and RS4;11 (**b**) cells treated with Dex, U73122, Edelfosine, 3-Nitrocoumarin, SPK-601, D609, and Manoalide alone or in combination. Cell mortality was determined by CCk-8 staining after 48 h of treatment. Data shown are the mean ± SEM of n = 3 (in **a**) and n = 3 (in **b**) independent experiments. Two-tailed, unpaired Student's t-test. **c** Nalm-6 cells treated with 100 nM Dex and 1 μM U73122 alone or in combination for 48 h. Alive and dead cells were determined by LIVE/DEAD® staining Kits using flow cytometry. **d** Quantification of alive and dead cell populations from (**c**). The percentage of cells was established after normalizing cells on Ctr cells. Mean ± SEM (n = 3 independent experiments), two-tailed, unpaired Student's t-test. **e** Nalm-6 cells treated with 100 nM Dex and 1 μM U73122 alone or in combination for 48 h. Cell mortality was determined by flow cytometry after annexin V/PI staining. **f** Quantification of total apoptotic cell populations from (**e**). Data are mean ± SEM (n = 3 independent experiments). Two-tailed, unpaired Student's t-test. **g** Cell viability of Dex-resistant primary diagnostic ALL blood (n = 16 patients) and bone marrow (n = 21 patients) samples exposed to 100 nM Dex and/or 1 μM U73122 for 24 h, normalized to Ctr condition. Data are mean ± SD. Two-tailed, unpaired Student's t-test. **h, i** Cell cycle analysis of Nalm-6 cells cells after stimulation with 100 nM Dex and/or 1 μM U73122 for 72 h. Cell cycle distribution (**h**) and subG1 quantification (**i**) were determined by PI staining followed by FACS analysis (n = 3 technical replicate). Two-tailed, unpaired Student's t-test. **j, k** Proliferation in Nalm-6 cells in the presence of 100 nM Dex and/or 1 μM U73122 for 3 days. Flow cytometry analysis showing CellTrace Violet (CTV) dilution (**j**) and proliferation quantification (**k**). Data are mean ± SEM (n = 3 independent experiments in **k**). Two-tailed, unpaired Student's t-test. **l** GSEA of RNA-Seq data showing enrichment in the cellular response to Dex stimulus comparing the transcriptome of Dex-resistant Nalm-6 cells treated with U73122 or Ctr (n = 2 biological replicates). See Supplementary Fig. 11c for differentially expressed gene profiles of cellular response to Dex stimulus-related genes. Source data are provided as a Source Data file.

Second, we used the intrinsically Dex-resistant B-ALL cell line RCH-ACV[11] and showed similar results using CXCR4 and PLC inhibitors (Fig. 8i). Taken together, these data support a model (Supplementary Fig. 21i) in which Dex, through activation of CXCR4 signaling, paradoxically induces its own resistance by activation of the PLC signalosome leading to apoptotic pathway braking, which in turn is sufficient to antagonize the proapoptotic effects of Dex.

## Discussion

Children with relapsed B-ALL become resistant to chemotherapy[4] with a low survival rate. Response to upfront GC treatment is a key element of therapeutic effectiveness, and its failure[5,6] is a major predictor of long-term B-ALL patient outcomes. Therefore, understanding the mechanisms underlying GC resistance is essential to prevent relapses and improve outcome. Given the prevalence of studies focused on understanding the resistance mechanisms observed during relapses[45,48], our objective was instead to decipher the mechanisms of the initial response to therapy responsible for primary or intrinsic GC resistance. In particular, the impact of the initial nongenomic GC response in B-ALL underscores a need to better understand the mechanisms governing how this response modulates GC sensitivity. Indeed, genomic pathway is not the only GC sensitivity factor, and not always correlated with GC sensitivity[3,7,43]. Ca²⁺ signaling is a versatile secondary messenger that regulates several cellular mechanisms, including cell growth and apoptosis[12–14]. We previously highlighted the importance of intracellular Ca²⁺ signaling in GC resistance in leukemia by showing that GC trigger a Ca²⁺-mediated pro-survival program, and that chelation of intracellular Ca²⁺ improves GC sensitivity[8]. Here, we show that PLC is an essential component of this GC-mediated intracellular Ca²⁺ signaling and B-ALL resistance to GC.

Our data support a mechanism in which GC paradoxically induce their own resistance by activation of CXCR4 signaling, leading to activation of the PLC signalosome. This ultimately results in the upregulation of cytosolic Ca²⁺ and the activation of PKC signaling. We here provide several lines of evidence which establish that CXCR4/PLC axis activity leads to the maintenance of B-ALL cell survival. Our finding that CXCR4 activated PLCγ2 signaling in B-ALL cells is consistent with the finding that PLCγ2-deficient B cells exhibited impaired SDF-1-signaling[16]. In the preclinical setting, CXCR4 inhibitors (such as plerixafor/AMD3100 or BKT140 and their derivatives) and genetic inhibition have been used as therapeutic agents to inhibit stroma-induced B-ALL cell growth/metabolism and overcome stroma-mediated drug resistance, and also, consistently with our results, to inhibit disease progression in mouse models of B-ALL[59,60]. In this light, it is therefore tempting to speculate that activation of CXCR4 signaling fulfills a pro-survival role in B-ALL cells, which is consistent with our results and also supported by previous works demonstrating that SDF-1 favors B-ALL cell survival and proliferation[34,60]. Our results demonstrate that Dex activates the CXCR4 receptor which manifests by internalization followed by intracellular Ca²⁺ release. A similar phenomenon has been reported in human T cells and eosinophils, in which GC upregulate CXCR4 expression[61–64] or modulate chemokine-mediated T cell redirection[65]. The exact mechanism by which GC/CXCR4 interaction influences T cell fate has not yet been deciphered, but it appears to modulate SDF-1-mediated signaling[66]. The present experiments show nongenomic mechanisms at the membrane level as the GC receptor antagonist (RU486) had no effect on Dex-mediated Ca²⁺ signaling. Furthermore, given the rapidity of the GC-induced CXCR4 internalization process, it cannot be excluded that GC act directly on CXCR4 since non-SDF-1 ligands have now been discovered[67] and also GC can directly bind to GPCR proteins as demonstrated recently[68].

One of the major conclusions of this study is the identification of PLC hyperactivation in GC-resistant cells. This constitutive pro-survival signaling is an important determinant that underlies the reliance of cancer B cells on ER Ca²⁺ store-mediated GC sensitivity. Interestingly, this is consistent with the finding that PLC is hyper-activated in prednisone-poor responder T-ALL patients accompanied by an increased GC resistance. In addition to mediating GC resistance in B-ALL, hypermorphic PLCγ2 mutations, which confer constitutive signaling have been shown to similarly induce drug resistance in chronic lymphocytic leukemia in which PLC signaling is important for survival and proliferation[18,20,22,23]. In contrast, PLC activation has been shown to be involved in GC-induced thymocyte apoptosis, suggesting a cell type-dependent action[69].

Our data suggest that the CXCR4/PLC axis-mediated GC resistance mechanism may also function in the presence of endogenous GC or in stromal cell-induced B-ALL cell resistance. Indeed, in co-culture with stromal cells, B-ALL cells exhibit substantial down-regulation of surface CXCR4, due to SDF-1-induced receptor internalization[59]. Based on the beneficial effects observed in mice, using PLC inhibitors may represent a treatment option in B-ALL. Although we did not observe overt toxicity in our mouse experiments or that PLCγ1 invalidation does not impair normal hematopoietic stem cells[21], the use of such pharmacological agents will have to be investigated with regard to their clinical benefit but also tolerability in patients. Altogether, the mechanism discovered here involving CXCR4 and PLC contributes to reduced susceptibility or intrinsic primary resistance of ALL to GC. This regulatory model provides a mechanistic explanation for the observations made 20 years ago[59,60] that CXCR4-mediated signaling has a negative effect on B-ALL sensitivity to GC. These observations support further investigation into the use of CXCR4 inhibitors and/or PLC modulators as a rational preventive strategy to limit the occurrence of resistance and improve the efficacy of Dex and chemotherapy in patients with B-ALL.

## Methods

All experiments were conducted following protocols that were approved by the Institutional Animal Care of Normandy University

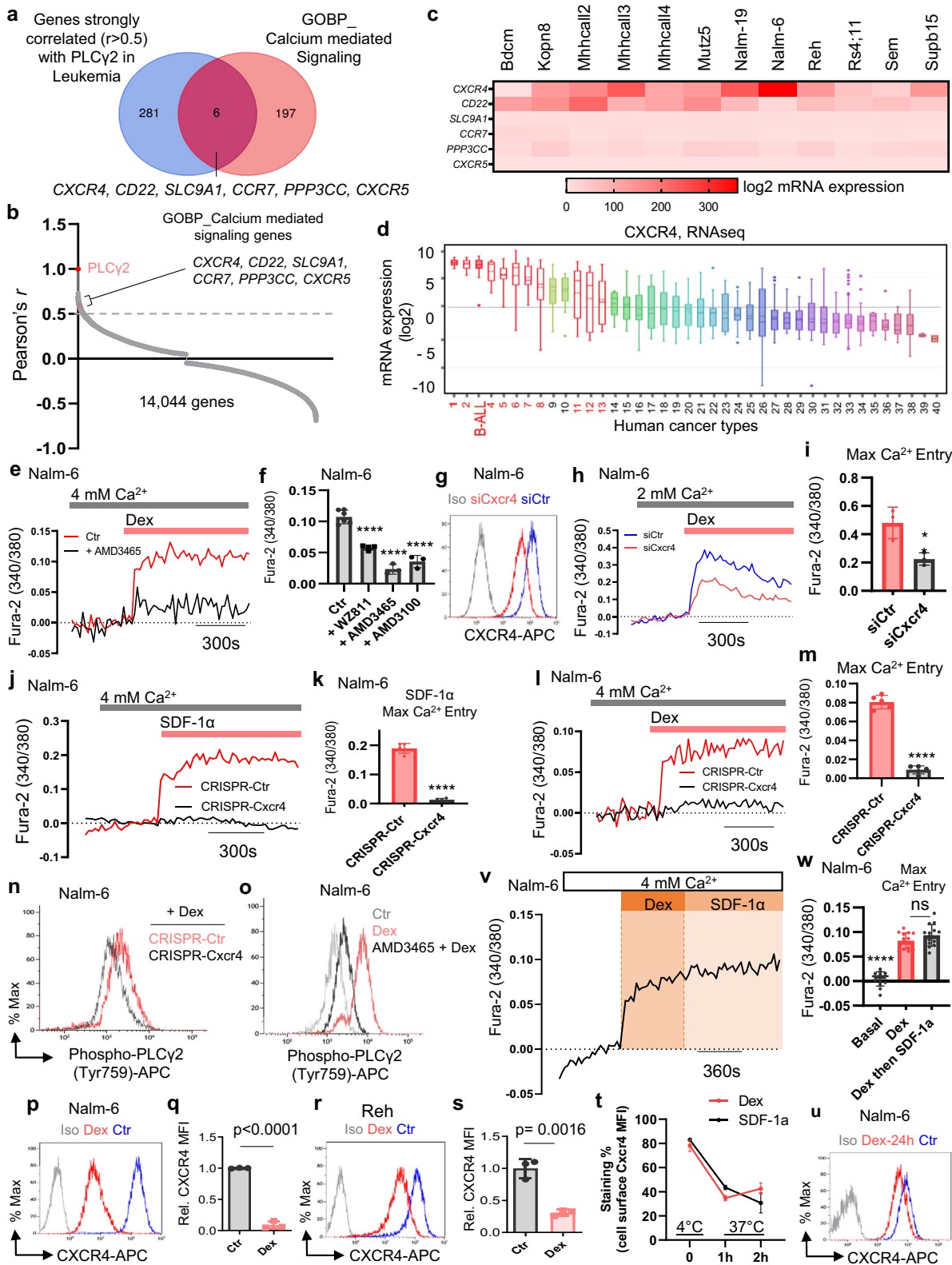

through the regional ethics committee for animal experimentation (CENOMEXA, APAFIS#27919). For ALL patient samples, protocols were approved by the institutional review board of Rouen University and Hospital Center (CHU Rouen), in accordance with the Declaration of Helsinki principles. This study is classified as non-interventional research by the CHU Research Project Qualification Commission (DRC-CQPR, 03/2020).

**In vivo mouse studies**

NOD.Cg-Prkdc^SCID Il2rg^tm1Wjl/SzJ (NOD SCID gamma, Charles River) mice were used at 6 to 12 weeks old male and female and were age-matched. NSG mice injected intravenously with sensitive or resistant Nalm-6 GFP/luciferase cells or RCH-ACV cells were treated after engraftment was established (luminescent flux assessed by biolumi-nescence imaging or by mCD45⁻ (1:200 dilution, clone 30-F11, BD

**Fig. 6 | Dex activates PLC/Ca2+ axis pathway in ALL cells through CXCR4.**
**a** Venn diagram depicting the overlap between genes strongly correlated ($r > 0.5$) with PLCγ2 in the MILE dataset versus $Ca^{2+}$ mediated signaling genes in GO biological processes. **b** Distribution of all MILE dataset genes based on their co-expression coefficient (Pearson's r) relative to PLCγ2. Genes linked to $Ca^{2+}$ mediated signaling highlighted in red are significantly enriched among the co-expressed genes. **c** Heatmap showing expression profiles of overlapping genes from (**a**) in different B-ALL cell lines. Data extracted from the cancer cell line encyclopedia (CCLE). **d** CXCR4 mRNA ranked by expression level in B-ALL and other human tumors. Data extracted from the CCLE. Numbers refer to cancer cell types with hematological malignancies depicted in red (see Supplementary Fig. 15a). Boxplots show the mean, median, and 75th to 25th percentiles. **e** Averaged cytosolic $Ca^{2+}$ measurements in Nalm-6 cells stimulated with 125 nM Dex with or without AMD3465. Data are mean ± SEM from ($n = 6$ for Ctr and $n = 3$ for antagonist; $n$-values correspond to independent experiments). **f** Quantification of maximal $Ca^{2+}$ entry of Nalm-6 cells stimulated with 125 nM Dex in the absence or presence of different CXCR4 antagonists. Data are the mean ± SEM from ($n = 6$ for Ctr, $n = 4$ for WZ811, and $n = 3$ for AMD3465 and AMD3100; $n$-values correspond to independent experiments). ****$p < 0.0001$. **g** Representative flow cytometry plots of CXCR4 surface expression in Nalm-6 cells transfected with small interfering RNA Ctr (siCtr) or siCXCR4. **h** Representative cytosolic $Ca^{2+}$ measurements in Nalm-6 cells transfected with either siCtr or siCXCR4 then stimulated with 125 nM Dex in nominally $Ca^{2+}$-containing buffer (2 mM $Ca^{2+}$). **i** Quantification of maximal $Ca^{2+}$ entry from (**h**). Data are mean ± SEM ($n = 3$ independent experiments, $p = 0.0209$). **j** Cytosolic $Ca^{2+}$ measurements in Nalm-6 cells transduced with either CRISPR-Ctr or CRISPR-Cxcr4 then stimulated with 100 nM SDF-1α in nominally $Ca^{2+}$-containing buffer (4 mM $Ca^{2+}$). **k** Quantification of maximal $Ca^{2+}$ entry from (**j**). Data are mean ± SEM ($n = 5$ independent experiments). ****$p < 0.0001$. **l** Cytosolic $Ca^{2+}$ measurements in Nalm-6 cells transduced with either CRISPR-Ctr or CRISPR-Cxcr4 then stimulated with 125 nM Dex in nominally $Ca^{2+}$-containing buffer (4 mM $Ca^{2+}$). **m** Quantification of maximal $Ca^{2+}$ entry from (**l**). Data are mean ± SEM ($n = 5$ independent experiments). ****$p < 0.0001$. **n, o** Representative flow cytometry plots of the active form of PLCγ2 (phospho-PLCγ2 (Tyr759)) induced by 125 nM Dex in Nalm-6 cells transfected with either CRISPR-Ctr or CRISPR-Cxcr4 (**n**) and in Nalm-6 cells in the absence or presence of AMD3465 (**o**). **p, r** Representative flow cytometry plots of CXCR4 surface expression of Nalm-6 cells stimulated with 125 nM Dex for 5 min at 37 °C. **q, s** Quantification of CXCR4 MFI from (**p, r**), respectively. Data normalized to Ctr are mean ± SEM ($n = 3$ independent experiments). **t** Nalm-6 cells were stimulated in a culture medium with either 100 nM Dex, 100 nM SDF-1α or medium alone. Cells were either transferred immediately on ice ($T = 0$) or after incubation for 1 or 2 h at 37 °C ($T = 1$ or 2 h) before staining with anti-human CXCR4 mAb. Values represent the percentage of staining, 100% corresponding to unstimulated cells processed in parallel. Data shown are the mean ± SEM ($n = 2$ for Dex, $n = 3$ for SDF-1α independent experiments). **u** Representative flow cytometry plots of CXCR4 surface expression of Nalm-6 cells stimulated with 125 nM Dex for 24 h. **v** Cytosolic $Ca^{2+}$ measurements in Nalm-6 cells stimulated with 125 nM Dex and 100 nM SDF-1α in buffer containing 4 mM $Ca^{2+}$. Data are mean ± SEM ($n = 3$ independent experiments). **w** Quantification of maximal $Ca^{2+}$ entry from (**v**). Data represent the mean ± SEM ($n = 7$ independent experiments, performed in duplicates). All statistical significance was analyzed by two-tailed, unpaired Student's $t$-test. *$p < 0.05$, and ****$p < 0.0001$. Source data are provided as a Source Data file.

---

Biosciences, 561018) and hCD19$^+$ (1:200 dilution, clone J3-119, Beckman Coulter, B49213) in mice peripheral blood). Mice were randomized into groups to equally distribute the leukemic burden (as assessed by bioluminescence or by flow cytometry) and weight in the different treatment groups. AMD3465 (dissolved in PBS) was administered subcutaneously (100 µL per mouse) at a concentration of 5 mg/kg for 3 days. U73122 (dissolved in DMSO then in PBS) was administered intraperitoneally (100 µL per mouse) at a concentration of 15 mg/kg for 3 days. The mice in the control and dexamethasone (Dex) only groups were treated with solvent as the second treatment. Dex (Enzo Life Sciences, catalog no. BML-EI126) was administered one hour after AMD3465/U73122 by intraperitoneal injection at a dose of 15–30 mg/kg/day in a cycle of 4–5 days on/3 days off during the 2 weeks of treatment. The endpoint for mice overall survival was reached when mice had their hind limbs paralyzed and the humane endpoint was not exceeded during our study. All mice were housed in individually ventilated cages in a barrier system under conditions of 12 h/12 h light/dark cycle at 20–25 °C and 45–65% of relative humidity.

### ALL patient samples
Patient B-ALL cells were obtained from the authorized research biological collection of the Department of Pediatric Immuno-Hemato-Oncology. Informed written consent was obtained from patients or their guardians at the time of sample collection. Primary blasts from 44 patients diagnosed with acute lymphoblastic leukemia, ALL, (peripheral blood and bone marrow) samples were used (Supplementary Table 1). Patients were treated with standard of care and patient characteristics, BM data, and laboratory values, including cytogenetics or phenotypes, were assessed before any treatment. Mononuclear cells were isolated by Ficoll-Paque Plus centrifugation. Cells were cryopreserved and stored in liquid nitrogen until use. After thawing, cells were maintained in RPMI 1640 medium supplemented as described below.

### Transduction of B-ALL cell line with luciferase/GFP
Nalm-6 cell line (ATCC cat. CRL-3273) was obtained from the American Type Culture Collection (ATCC). Cells were authenticated and tested for mycoplasma. To track leukemia burden with Bioluminescent

imaging (BLI) in vivo model, Nalm-6-BLIV cells expressing firefly luciferase and green fluorescent protein (GFP) were established by transducing MSC-GFP-T2A-Luciferase lenti-reporter vector (System Biosciences, cat. BLIV301PA-1) into Nalm-6 cells with calcium phosphate transfection method.

### Intracellular $Ca^{2+}$ measurement
ALL cells were labeled with Fura-2 QBT (Molecular Devices, R8198, UK) for 60 min at 37 °C in buffer containing (in mM) 5KCl, 135NaCl, 1MgCl$_2$, 2-4CaCl$_2$, 10 HEPES, 10glucose, pH 7.4 in the presence or absence of reagents and/or inhibitors on Cell-Tak (Corning, NY) precoated 96-well plates. After 100 s of recording, cells were stimulated with test compounds. Store depletion was induced by stimulating the cells with 2 µM thapsigargin (Invitrogen) in $Ca^{2+}$-free buffer and $Ca^{2+}$ influx was induced after adding of 2 or 4 mM $Ca^{2+}$ buffer to the cells at 750 s. Cytosolic and SOCE $Ca^{2+}$ entry were quantified by the peak of the F340/F380 ratio. For experiments in $Ca^{2+}$-free buffer, CaCl$_2$ was replaced by EGTA (2 mM). Changes in Fura-2 ratios (F340/380) were measured using a FlexStation 3 Multi-Mode Microplate Reader (Molecular Devices) and analyzed using the SoftMax Pro 7.1 software (Molecular Devices). For some experiments, $Ca^{2+}$ was recorded under Leica DMI6000 B inverted microscope (Leica, Nanterre, France) equipped with SENSICAM EM camera for real-time recording as described above.

### CRISPR/Cas9 targeting CXCR4 in ALL cell lines
Nalm-6 and RS4;11 cells expressing CXCR4 were transfected in ultra-cruz transfection reagent (sc-395739, Santa Cruz Biotechnology, Dallas, TX, USA) either with human CXCR4-targeted CRISPR/Cas9 knockout plasmid (CXCR4 CRISPR/Cas9 KO Plasmid (h), sc-400254) or non-specific Control CRISPR/Cas9 plasmid (sc-418922; Santa Cruz Biotechnology) according to the manufacturer's instructions. All plasmids encoded a GFP marker to indicate transfection. Successful transfections were confirmed visually by detection of GFP under fluorescent microscopy according to the manufacturer's instructions. ALL cells were further sorted after the transfection step to eliminate GFP-negative cells. For this, cells were resuspended in fetal bovine serum (FBS) and cell sorting was performed using a FACSAria cell sorter (BD Biosciences). Sorted cells were denoted CRISPR-Ctr cells

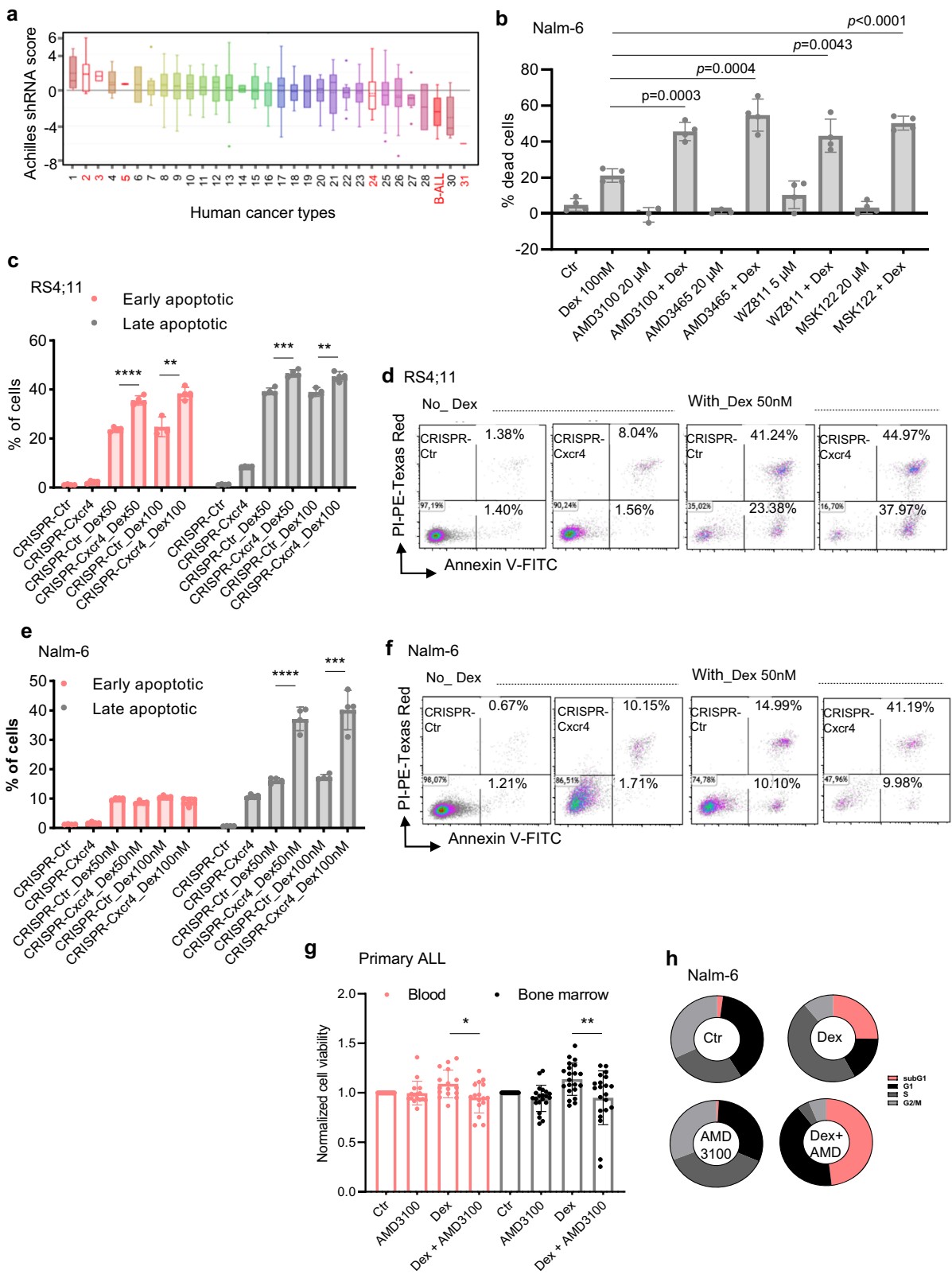

and CRISPR-Cxcr4 cells, respectively, plated and cultured in standard growth medium before the experiments.

## B cell isolation and culture

Human B cells from healthy volunteers obtained via the French blood bank were separated from whole blood by density gradient centrifugation using Ficoll-Paque Plus (GE Healthcare) and Dynabeads

Untouched Human B cells kit (Life Technologies) following the manufacturer's instructions. Cell purity was assessed by flow cytometry analysis and was over 95% for B cells (CD19+). B cells were cultured in RPMI 1640 supplemented with 10% fetal bovine serum, 2 mM L-glutamine, 100 U/ml penicillin, and 100 µg/ml streptomycin (all reagents from Eurobio®) at 37 °C with 5% $CO_2$.

**Fig. 7 | CXCR4 inhibition enhances Dex sensitivity in ALL cell lines in vitro and ex vivo. a** CXCR4 shRNA ranked by Achilles score level in B-ALL and other human tumors. Data were extracted from the CCLE. Numbers refer to cancer cell types (see Supplementary Fig. 20a) among which hematological malignancies are depicted in red. Boxplots show the mean, median, and 75th–25th percentiles. **b** Nalm-6 cells were treated with Dex in combination with different CXCR4 inhibitors. Cell mortality was determined by CCk-8 staining after 48 h. The percentage of dead cells was established after normalizing cells on Ctr cells. Data shown are the mean ± SEM ($n = 4$ independent experiments). Two-tailed, unpaired Student's $t$-test. **c, e** Cell mortality of CRISPR/Cas9 CXCR4-invalidated RS4,11 and Nalm-6 cells treated with Dex (nM) for 48 h determined by flow cytometry after annexin V/PI staining. The percentage of early and late apoptotic cells was calculated by the percentage of annexin V-FITC positive and annexin V-FITC positive plus annexin V-FITC/PI-positive population, respectively. Data are the mean ± SEM ($n = 4$ independent experiments). Two-tailed, unpaired Student's $t$-test. **c** ****$p < 0.0001$, **$p = 0.0027$ (early); ***$p = 0.0004$, **$p = 0.0069$ (late). **e** ****$p < 0.0001$, **$p = 0.0005$ (late). **d, f** Representative flow cytometry plots from (**c, e**). **g** Viability of blood ($n = 16$ patients) and bone marrow ($n = 21$ patients) cells from 37 Dex-resistant primary diagnostic ALL patient samples (see Supplementary Fig. 10b) exposed to 100 nM Dex and/or 25 μM AMD3100 for 24 h, normalized to Ctr condition, Statistical significance was determined by two-tailed, unpaired Student's $t$-test. Data are the mean ± SD. *$p = 0.0136$ (blood), **$p = 0.0095$ (bone marrow). The experiment was carried out under the same conditions as in Supplementary Fig. 10b. **h** Cell cycle analysis of Nalm-6 cells after stimulation with 100 nM Dex and/or 25 μM AMD3100 for 48 h. Cell cycle distribution determined by flow cytometry after PI staining. Statistical significance was determined by two-tailed, unpaired Student's $t$-test. *$p < 0.05$, **$p < 0.01$, ***$p < 0.001$ and ****$p < 0.0001$. Source data are provided as a Source Data file.

## Bioluminescent imaging

NSG mice were imaged with the Vilber Smart In Vivo Imaging System (Vilber lourmat). Mice were sedated with 2.5% isoflurane, followed by subcutaneous XenoLight D-Luciferin-K+Salt Bioluminescent Substrate (Elmer Perkin) injection and subsequent imaging. Quantification and image processing were performed using Newton 7.0 Software (Vilber smart Imaging).

## Cell lines

The B-ALL cell lines Nalm-6, RS4;11, Reh, HAL-01, and RCH-ACV (Deutsche Sammlung von Mikroorganismen und Zellkulturen, DSMZ®, Braunschweig, Deutschland) were cultured in RPMI 1640 medium (Eurobio®, Courtaboeuf, France) containing 10% fetal bovine serum (FBS, Eurobio®), 2 mM of L-glutamine (Eurobio®) with 5000 UI/l penicillin and 50 mg/l streptomycin (Eurobio®). B-ALL cell lines were maintained at 37 °C in a 5% CO2 humidified atmosphere. All cell lines were authenticated via short tandem repeat (STR) methodology and routinely tested for mycoplasma contamination using the MycoAlert PLUS detection kit (Lonza, LT07-705).

## Calcium live-cell imaging

ALL cells were cultured on Krystal 24-well glass-bottom plates (Proteigene, Saint-Marcel, France) coated with poly-D-lysine and then incubated with Fura-2/AM (5 μM) diluted in culture medium for 60 min at 37 °C. After loading, cells were washed three times (600 g × 10 min) and remained suspended in a buffer solution containing: 110 mM, NaCl; 5.4 mM, KCl; 25 mM, NaHCO$_3$; 0.8 mM, MgCl$_2$; 0.4 mM, KH$_2$PO$_4$; 20 mM, Hepes; 0.33 mM, Na$_2$HPO$_4$; 1.2 mM, CaCl$_2$, pH adjusted to 7.4. For experiments in Ca$^{2+}$-free medium, CaCl$_2$ was replaced by EGTA (2 mM). Fura-2/AM was excited with 340 nm and 380 nm laser lines, and the emitted fluorescence was recorded at 510 nm under the Leica DMI6000 B inverted microscope (Leica, Nanterre, France) equipped with SENSICAM EM camera for real-time recording of fluorescent images. F340/F380 emission ratios were calculated for each time point.

## Generation of Dex-resistant B-ALL cell line

To establish Dex-resistant cells, we treated the Nalm-6 cell line with gradually increasing Dex concentrations from 0 to 5 μM over two months. The culture medium was changed every 3–4 days, and the Dex doses were doubled when the treated cells started to proliferate at an equal rate to the untreated parental cells. The resistant cells (Nalm-6 R) were further grown for 2 weeks in Dex-free culture medium to the end of the protocol. To determine drug sensitivity, resistant or control (DMSO) cells were seeded in 96-well plates for 24, 48, 72 h in the presence of different concentrations of Dex (from 0 to 100 μM) and CCK-8 assay was performed as described elsewhere.

## MTT Assays

Cells were seeded in 96-well culture plates with or without test compounds. After 48 h, culture medium was replaced by 100 μl serum free medium and 10 μl MTT (3-(4,5-dimethylthiazol-2-yl)-2,5-diphenyltetrazolium bromide) solution (5 mg/ml) were added to each well, and the plates were incubated at 37 °C for 3 h. The medium was aspirated and 50 μl DMSO was added to each well to dissolve the formazan crystals. After 15 min at 37 °C, the absorbance was read at 540 nm using a Spectrofluorometer SAFAS Xenius XC (MC 98000 Monaco) and the SAFAS SP2000 program. Alternatively, Cell viability control was assessed by the trypan blue exclusion test, which stains dead cells blue.

## CCK-8 assays

Cultured cells were seeded in 96-well culture plates with or without test compounds. After 48 h, cell viability was assessed by using cell counting kit-8 (CCK-8, Sigma, France) according to the manufacturer's instructions, and the absorbance was read at 450 nm under Flexstation 3 Molecular Devices (UK).

## LIVE/DEAD® Viability/Cytotoxicity Kit assay

Cultured cells were treated with or without inhibitors and Dex for 48 h. Then, cell viability was assessed by using LIVE/DEAD® Cell Vitality Assay (Abcam, UK). The kit is a mixture of two compounds: (i) calcein AM, a fluorogenic esterase substrate which gives green fluorescent product when hydrolyzed, indicating that cells sustain esterase activity and that their membranes are not damaged, and (ii) ethidium homodimer-1, a red-fluorescent nucleic acid fluorophore that can penetrate the cells that have impaired membranes. Subsequently, live cells are stained in green while dead cells appear red. Cells were incubated with the reagents in the LIVE/DEAD® kit for 10 min at 37 °C and samples were analyzed on LSRFortessa (BD Biosciences) flow cytometry cell analyzer and analyzed using Kaluza software (kaluza A85810 AB).

## Proliferation assay

CellTrace™ Violet (CTV) Cell Proliferation (Life Technologies, USA) assay was used to assess cell proliferation through flow cytometry. Cells were labeled with CellTrace™ Violet dye (1:1000 PBS) for 10 min, protected from light at room temperature, and resuspended in pre-warmed complete culture medium in the presence of inhibitors and/or Dex. CTV dilution was assessed by flow cytometry 72 h after stimulation.

## Cell cycle analysis

ALL cells were treated with or without test compounds alone or in combination for 72 or 96 h and then washed with cold PBS and fixed with cold 80% ethanol for 30 min at RT. After washing, cells were incubated for 15 min (in the dark) with 1 mg/ml RNase A, 1 mg/ml propidium iodide (PI) in PBS. Cell cycle distribution was acquired on LSRFortessa cell analyzer (BD Biosciences) and analyzed using Kaluza software (kaluza A85810 AB).

## Apoptosis assays

Apoptotic cells were determined by Annexin V-FITC staining (Abcam Biochemicals (Paris, France)). Annexin V-FITC and/or PI-positive cells

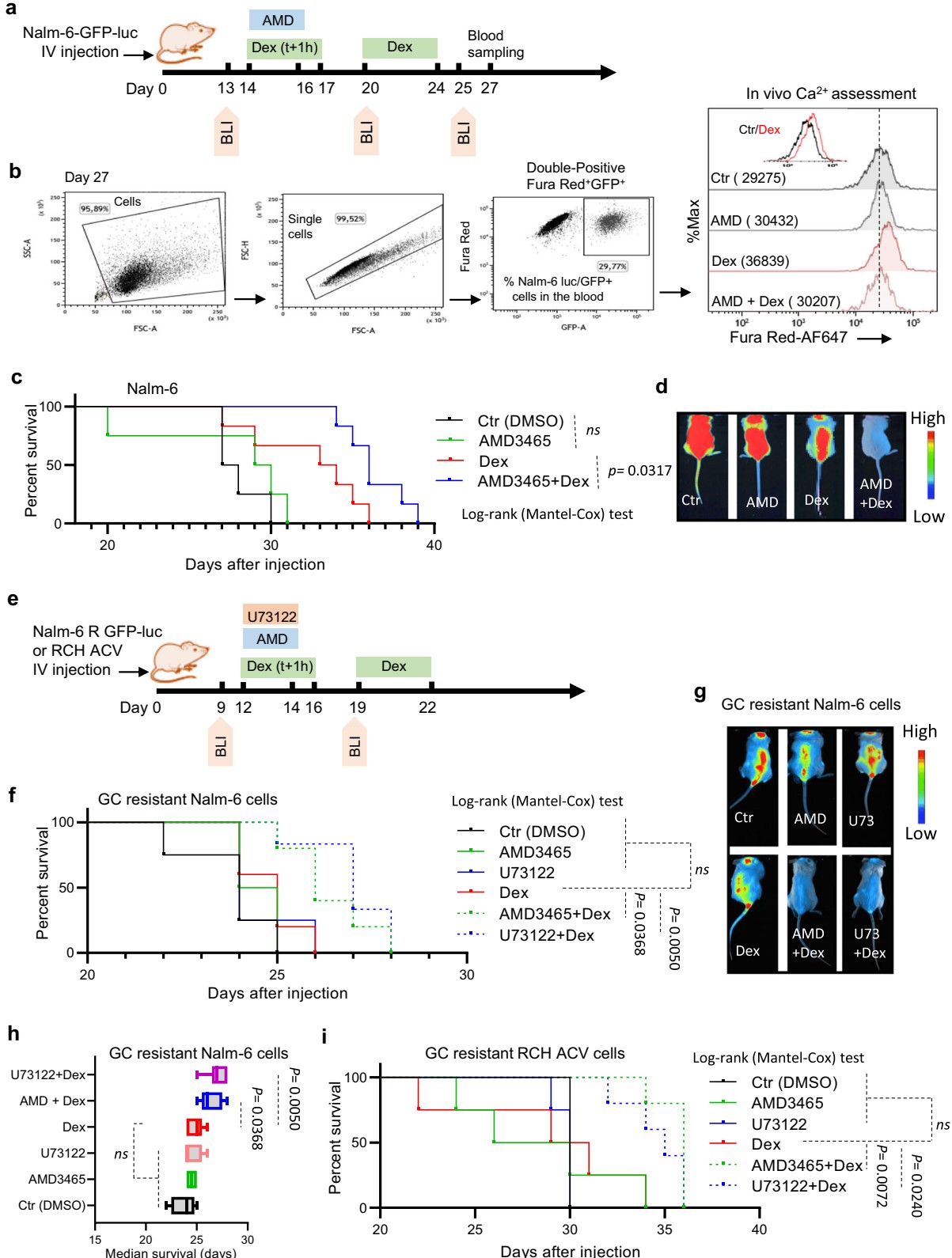

were assessed by flow cytometry using BD FACSCanto II Flow Cytometer (BD Biosciences, USA) and analyzed using Kaluza software.

### Caspase-3 Activity

Activity of caspase-3 was measured using Ac-DEVD-AFC substrate (Enzo Life Sciences (ELS) AG, Villeurbanne, France), according to the manufacturer's instruction. Briefly, ALL cell lines were treated with or without test compounds for 24 h, and then collected and lysed in cell lysis buffer (HEPES 50 mM, NaCl 100 mM, DTT 10 mM, CHAPS 0.1%, EDTA 1 mM, pH 7.4). Cell lysate (20 µg) was added to 100 µL of caspase-3 buffer containing 40 µM of the caspase-3 substrate Ac-DEVD-AFC (fluorogenic) as final concentration and incubated at 37 °C for 1 h in the dark. Caspase-3 activity was assessed by measuring fluorescence (Ex 400 nm/Em 505 nm) using SAFAS Xenius XC

**Fig. 8 | CXCR4/PLC inhibition improves survival of Dex-treated NSG mice in vivo. a** Experimental design of NSG mice treatment experiments. BLI bioluminescence imaging. **b** Flow cytometry Ca$^{2+}$ assessment in Nalm-6 GFP$^+$ cells in the blood at day 27. **c** Kaplan–Meier survival analysis of mice transplanted with Nalm-6 GFP-luc cells and treated with Ctr ($n = 4$), Dex ($n = 6$), AMD3465 ($n = 4$), or a combination of Dex and AMD3465 ($n = 6$), $n$-values correspond to individual mice. Statistical significance was calculated with the Log-rank (Mantel-Cox) test. **d** An example of bioluminescence imaging of Nalm-6 GFP/luc at day 25 of tumor challenge from (**c**). **e** Experimental design of NSG mice treatment experiments using Dex-resistant B-ALL models. BLI bioluminescence imaging. **f** Kaplan–Meier survival analysis of mice transplanted with resistant (R) Nalm-6 GFP-luc cells and treated with Ctr ($n = 4$), Dex ($n = 5$), AMD3465 ($n = 4$), U73122 ($n = 4$) or a combination of Dex and AMD3465 ($n = 5$) or and U73122 ($n = 6$), $n$-values correspond to individual mice. Statistical significance was calculated with the Log-rank (Mantel-Cox) test. **g** An example of bioluminescence imaging of Nalm-6 R GFP/luc at day 19 of tumor challenge from (**f**). **h** Averaged of the median survival over time is shown for the different treatment groups from (**f**). Data are the mean ± SEM from ($n = 4$ for Ctr, AMD3465, U73122; $n = 5$ for Dex and Dex+AMD and $n = 6$ for Dex+U73122; $n$-values correspond to individual mice. Boxplots show the mean, median, and 75th–25th percentiles. **i** Kaplan–Meier survival analysis of mice transplanted with resistant RCH-ACV cells and treated with Ctr ($n = 4$), Dex ($n = 4$), AMD3465 ($n = 4$), U73122 ($n = 4$) or a combination of Dex and AMD3465 ($n = 5$) or and U73122 ($n = 5$). Statistical significance was calculated with the Log-rank (Mantel-Cox) test. Source data are provided as a Source Data file.

Spectrofluorometer (MC 98000 Monaco) or Flexstation 3 Molecular Devices (UK).

### Flow cytometry Ca$^{2+}$ measurement
Cytosolic Ca$^{2+}$ levels were monitored with two cytosolic Ca$^{2+}$ indicators Fluo-4 AM and Fura Red™, AM. B-ALL cells were loaded for 30 min with 2 μM Fluo-4 AM or Fura Red at room temperature in buffer described above, followed by a de-esterification step of 30 min in the absence of probes. During the de-esterification step, cells were treated with compounds. Flow cytometry data were collected on a BD LSRFortessa, and analyzed with Kaluza software. Alternatively, fluorescence was also monitored on a luminescence FlexStation 3 Multi-Mode Microplate Reader (Molecular Devices) and analyzed using the SoftMax Pro 7.1 software (Molecular Devices).

### Phospho flow cytometry
The measurement of PLCγ2 protein phosphorylation following Dex stimulation was performed as previously described[7]. Briefly, ALL cells were treated with or without CXCR4 antagonists for 30 min and then stimulated with Ctr or 125 nM Dex for 5 min. Cells were subsequently fixed with fixation buffer (Biolegend, cat. no. 420801) and permeabilized with methanol. PLCγ2 protein phosphorylation was assessed using APC anti-Phospho-PLCγ2 (Tyr759) (5:100 dilution, clone 4NPRN4, ThermoFisher, eBioscience, 17-9866-42), PE anti-Phospho-PLC-γ2 (pY759) (2:10 dilution, clone K86-689.37, BD Biosciences, BD558490) or PE Mouse IgG1, kappa Isotype control (1:200 dilution, clone MOPC-21, Biolegend, 400111). For the effect of inhibitors on basal PLCγ2 phospho level, ALL cells were exposed to Ctr or inhibitors for 24 h. Flow cytometry data were collected on a BD LSRFortessa and analyzed with Kaluza software.

### Measurement of GR, PLC levels by flow cytometry
ALL cells were stimulated with or without test compounds, and levels of glucocorticoid receptor (GR) and total phospholipase C (PLC) were assessed by intracellular staining flow cytometry[70]. After fixation with buffer (Biolegend, cat. no. 420801) and permeabilized with True-Phos Perm Buffer (Biolegend, cat. no. 425401), cells were stained FITC anti-GR (5:100 dilution, clone G-5, Santa Cruz Biotechnology, sc-393232), Alexa Fluor® 647 anti-PLCγ2 (5:100 dilution, clone B-10, Santa Cruz Biotechnology, sc-5283), FITC anti-PLC γ1 (5:100 dilution, clone E-12, Santa Cruz Biotechnology, sc-7290). The complete list of antibodies used in this study can be found in Supplementary Table 2. Flow cytometry data were collected on a BD LSRFortessa, and analyzed with Kaluza software.

### PLC enzyme activity
ALL cells were treated with or without test compounds alone or in combination for 24 h. The measurement of phospholipase C activity was assessed using the EnzChek™ Direct Phospholipase C Assay Kit (Invitrogen, no: E10215) according to the manufacturer's instructions.

### PKC kinase activity assay
ALL cells were treated with or without test compounds alone or in combination for 24 h. The measurement of PKC kinase activity was assessed using the PKC Kinase Activity Assay Kit (Abcam Biochemicals, no: ab139437,) according to the manufacturer's instructions. Alternatively, PKC kinase activity was assessed using the PKC Kinase Activity Kit (Enzo Life Sciences, no: ADI-EKS-420A) according to the manufacturer's instructions.

### Short interfering RNA knockdown
Silencing experiments were performed with ON-TARGETplus Human siRNA (25 nM) targeting CXCR4 (Horizon-Dharmacon, J-005139-08) or non-targeting siRNA as a control (Horizon-Dharmacon, D-001810-10). siRNAs were introduced into ALL cells using DharmaFECT Transfection Reagent (Dharmacon, T-2001) according to the manufacturer's instructions. Silencing experiments were also performed with Human siRNA (25 nM) targeting PLCγ1 and PLCγ2 (Santa Cruz Biotechnology, sc-29452 and sc-36268) or control siRNA-A (Santa Cruz Biotechnology, sc-37007). In this case, siRNAs were introduced into ALL cells using siRNA Transfection Reagent (Santa Cruz Biotechnology, sc-29528) according to the manufacturer's instructions. After 24-h incubation, the medium was aspirated and replaced with fresh medium without siRNA for 48 h. Transfected ALL cells were then used for the Ca$^{2+}$ signaling analysis or other experiments.

### Measurement of cell surface CXCR4
For analysis of cell surface CXCR4 expression, ALL cells were treated with or without test compounds alone or in combination at times indicated in the legends. Antibodies against human (5:100 dilution, clone 12G5, Ebioscience, 17-9999-42), (5:100 dilution, clone 12G5, Biolegend, 306510) or (5:100 dilution, clone 12G5, BD Biosciences, BDB555976) CXCR4-APC were used. Data are presented as the median fluorescent intensity (MFI) of the CXCR4 signal.

### CXCR4 receptor internalization analysis
CXCR4 receptor internalization was performed as previously described[54]. Briefly, ALL cells were maintained in RPMI medium as described above. They were then stimulated with or without medium supplemented with 125 nM Dex or 100 nM SDF-1α, and kept either at 4 °C (for $T = 0$) or incubated at 37 °C for the times indicated in the figures. All subsequent steps were carried out at 4 °C. Cells were washed once in staining FACS buffer (PBS, 0.5% BSA, 0.05% NaN3, and 5% FBS) and incubated in the presence of anti-human CXCR4-APC (5:200 dilution, clone 12G5, eBioscience, France, cat. no. 17-9999-42) or isotype control APC-Mousse IgG 1k (1:200 dilution, clone MOPC-21, Biolegend, cat. no. 400119) / isotype control APC-Mousse IgG 1k (1:200 dilution, clone MOPC-21, BD Biosciences, cat. no. 554681) for 30 min. After two washes, signals were acquired on flow cytometer. Results are given as percentage of MFI controls, 100% corresponding to unstimulated (medium alone) cells processed in parallel. In some cases, Ca$^{2+}$ signaling was also measured under the same conditions.

## RNA sequencing

Nalm-6 R cells were treated with or without PLC inhibitor, U73122, for 16 h. Total RNAs were then extracted with NucleoSpin RNA kit according to the manufacturer's recommendations (Macherey-Nagel) and stored at −80 °C until use. The quality and quantity of isolated mRNAs were assessed using 2100 Bioanalyzer (Agilent Technologies, Santa Clara, CA, USA) and the Qubit 3.0 device (Thermo Scientific, Wilmington, USA). Only RNA samples with a minimal RNA integrity number of 8 were used for library preparation. RNA-Seq libraries were prepared using the NEBNext Ultra II Directional RNA Library Kit for Illumina (New England Biolabs, Ipswich, USA) kit, and High-throughput sequencing of the libraries was performed on an Illumina NextSeq500/550 (Illumina, San Diego, USA) using 2*75 bp sequencing to generate 30 M read pairs on average per sample. Bioinformatics analysis was carried out using nf-core/rnaseq v3.1 analysis pipeline (https://nf-co.re/rnaseq) to generate multi quality control report that uses the STAR v2.6.1d and SALMON v1.4.0 tools for alignment. Differential expression analysis, based on a model using the binomial negative distribution, was performed with DESeq2 tool to evaluate significant counts in cells treated with U73122 compared to untreated (control) cells.

Additional RNA-Seq data were extracted from the Gene Expression Omnibus (GEO) database and are listed in the Data Availability section.

## Gene set enrichment analysis (GSEA)

Raw data in two biological replicates were normalized for GSEA analysis. Genes were ranked according to the degree of differential expression, and then the predefined gene set was analyzed to determine which were enriched at the top or bottom of the list. Gene expression signatures and canonical pathway analyses were performed using the GSEA analysis tool http://www.broadinstitute.org/gsea/index.jsp, and several sets (GO, KEGG, Reactome, Biocarta, and Wikipathways) sets were used for GSEA independently.

## Reagents

Source data for each reagent (chemical, materials, critical commercial assays, sequencing reagents, experimental models, antibodies, transfection, and plasmids) used in this paper are provided in Supplementary Table 2 with information including reagents' full names, manufacturer's name, and ID.

## Reproducibility and statistical analysis

Data are expressed as mean ± S.E.M. values from at least duplicate independent experiments. Statistical analyses were conducted using GraphPad PRISM Software (v 8.0.2) using two-way analysis of variance (ANOVA) with Sidak's multiple comparisons test, two-tailed unpaired Student's $t$-test, two-tailed paired $t$-test, and two-tailed Mann-Whitney test, as mentioned in the figure legends. No statistical method was used to predetermine the sample size. No data were excluded from the analyses. Mice were randomized into different groups to equally distribute the leukemic burden (as assessed by bioluminescence). The investigators were not blinded to outcome assessments. Differences were considered statistically significant at $p < 0.05$. For survival analysis, the log-rank test was used to perform pairwise comparisons between survival curves.

## Reporting summary

Further information on research design is available in the Nature Portfolio Reporting Summary linked to this article.

## Data availability

The RNA-Seq data generated for this study have been deposited in the GEO database under accession number GSE214990. The calcium mediated signaling genes were downloaded from the Molecular Signatures Database (MSigDB, http://www.gsea-msigdb.org/gsea/msigdb/index.jsp). Additional available gene expression datasets used in this study were downloaded from GEO or ArrayExpress (https://www.ebi.ac.uk/biostudies/arrayexpress) under the following accession numbers: GSE655 and GSE656 from GEO for gene expression analysis of primary B-ALL sensitive and resistant to prednisolone[44], respectively. GSE5820 from GEO for gene expression analysis of primary ALL sensitive or resistant to glucocorticoids[45]. GSE28460 and GSE18497 from GEO for array expression data from patients at diagnosis and relapse[46,47]. E-MTAB-7781 from ArrayExpress for RNA-seq data[43] (https://www.ebi.ac.uk). GSE13159, GSE63157, GSE45547, GSE37642 and GSE10846 for RNA-seq or survival data from the R2 database (http://r2.amc.nl)[35,71-74]. GSE13204 for microarray data[37] from Oncomine[38]. GSE7186 for microarray data from GEO[52]. Additionally, published gene expressions analyzed in this study were downloaded from Cancer Cell Line Encyclopedia (CCLE)[36], and Gene Expression Profiling Interactive Analysis (GEPIA)[39]. The remaining data are available within the Article, Supplementary Information or Source Data file. Source data for each figure and supplemental figure are provided with this paper. Source data are provided with this paper.

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

## Acknowledgements

This study was supported by grants from "Ligue Contre le Cancer" (AK/IP/BC/16094/16792), France, the Comité Départemental de la Ligue Contre le Cancer, Comité de Loire-Atlantique (IP/BC-17040), France to S.A.A. and Association "Vie et Espoir", France to S.A.A. We thank Sophie Coutant (Inserm U1245) for her technical assistance in RNA sequencing. We are grateful to Nikki Sabourin-Gibbs (CHU Rouen) for her help in editing the manuscript.

## Author contributions

S.A.A. conceived the study, performed the experiments, analyzed and interpreted the data, prepared figures, and wrote, revised, and reviewed the manuscript. S.A.A. and R.H. performed bioinformatic analysis. S.A.A. and G.R. performed in vivo experiments. S.A.A., C.D., and C.C. performed RNA-Seq experiments. J.P.V., M.L.G., and P.S. reviewed the manuscript. O.B. analyzed the data and reviewed the manuscript.

## Competing interests

S.A.A., J.P.V., and O.B. are designated as inventors for the European Patent application EP23307352 filed on December 22nd, 2023 in the names of Inserm, Université de Rouen, and CHU de Rouen and entitled "methods for preventing resistance to chemotherapy in acute lymphoblastic leukemia". The remaining authors declare that the research was conducted in the absence of any commercial or financial relationships that could be construed as a potential conflict of interest. OB received research funding and/or honoraria from Argenx, BMS, CSL Behring, Egle Tx, OGD2, and UCB. M.G. received research funding from Bridge-Medicines and holds equity of SeqRX. All other coauthors declare no competing interests.
