## [Peer Review File · Nature Communications]

Reviewers' Comments:

Reviewer #1:

Remarks to the Author:

The manuscript by Abdoul-Azize et al. investigates PLC/CXCR4 mediated glucocorticoid resistance in B cell ALL. The authors show that dexamethasone treatment results in CXCR4 internalization, activation of PLC and results in therapy resistance. Treatment with PLC inhibitors or CXCR4 antagonists sensitize B ALL cells to dexamethasone. Overall, the manuscript is very dense and in some parts a challenge to follow. The manuscript would be significantly improved by increasing the clarity of the results section and figures. Additional scientific comments are below.

1. The authors explore many different aspects of biology in this manuscript including cell cycle, transcriptional changes, mitochondrial biology, ROS production, Ca²⁺ signaling, and CXCR4. The addition of rescue experiments to show that the changes observed are critical to the mechanism being described would improve the study overall. For example, does antioxidant treatment rescue cell death showing ROS production is a key part of the mechanism?

2. Are there any changes in PLC gamma, or CXCR4 expression from diagnosis to relapse in B ALL?

3. For combination treatments, it would be helpful to show the effect of vehicle, both single agents, and combination therapy to see if the effect of the combination treatment is greater than single agents alone. The same applies to the RNAi/CRISPR experiments. Showing control, siRNA or CRISPR cells with and without dexamethasone is essential to understand if the biology observed is due to CXCR4 knockdown/out alone or the loss of CXCR4 with dexamethasone.

4. Many of the experiments appear to have been performed with GC sensitive cell lines. While it is appreciated that some experiments were done in resistant lines, it would be helpful to repeat experiments in GC resistant lines (either intrinsically resistant or the authors Nalm6 dex-resistant line) to see if the mechanism is the same in dexamethasone resistant ALL. This would be particularly important to do for the mouse experiments.

Reviewer #2:

Remarks to the Author:

In their manuscript, Abdoul-Azize and colleagues describe glucocorticoid-triggered CXCR4 signaling as an activator of the PLC signalosome that may promote tumor resistance in B cell acute lymphoblastic leukemia. This is a timely and interesting topic and the authors have shown their expertise investigating signaling in hematologic neoplasms and immune responses.

Here, they describe a relevant signaling axis, that may contribute to resistance in B-ALL. While the exploration of the signaling pathway is detailed and conclusive, the manuscript falls short in relevant aspects, that should be addressed to improve the quality of the already impressive manuscript.

Major aspects:

- The authors indicate that PLCg2 is highly expressed in B-ALL on the mRNA level. The mechanism behind this aberrant induction of PLCg2 expression remains unexplained. Is the induction of PLCg2 expression influenced by the underlying oncogene? Is the expression heterogeneous in samples with different driver mutations? The reason for high PLCg2 expression specifically in B-ALL should be mechanistically explained, e.g. by using reporter systems for PLCg2 in combination with genetic or pharmacologic screening approaches.

- While the authors focus on PLCs as mediators of Ca⁺⁺ signaling and the role of PLC family members, they do not mention or investigate the role of PLCg2's human homologue PLCg1. Although not highly expressed, it may have non-redundant functions in B-ALL (both on Ca⁺⁺ signaling and for activation of other signaling pathways). PLCg1 could partially compensate for its homologue. Inactivation of PLCg1 in B-ALL cell lines followed by functional analyses and investigation of Ca⁺⁺ signaling should be implemented as a relevant control. Also, specificity of PLCg2 inhibitors used (especially regarding the activity of PLCg1) need to be shown.

- Functional analysis in vivo (Figure 8f) is restricted to NALM6-cells injected into mice. This data

requires corroboration by the use of patient-derived xenografts. Several models of B-ALL xenografts are available and published. At least 2 different B-ALL xenografts (with different underlying genetics) should be for treatment with Dex, AMD3465 and the combination versus control.

- While the authors provide evidence for the activation of the CXCR4-PLCg2-Ca⁺⁺ signaling axis, functional perturbations to assess for further mechanistic exploration are lacking. Genetic screening in PLCg2 deficient (e.g. by pharmacologic or genetic inactivation) cell lines would further confirm the relevance of the proposed signaling axis and be helpful to identify relevant interactors (or unknown intermediates). This experiment could also be performed in GC resistant cell lines (versus sensitive) to validate the signaling axis.

- Investigation of Ca⁺⁺ signaling does not include relevant controls: Fura-2 measurements in cell lines should be controlled by a second assay. Intracellular Ca⁺⁺ sensors and assessment by microscopy could serve as an appropriate control. Likewise, it is not described in this manuscript whether CXCR4-PLCg2 signaling initiates a STIM and Orai dependent calcium signals. It has been demonstrated in primary and leukemic T and B lymphocytes, that hyperactivation of SOCE can drive a hyper-proliferative state. This also emphasizes the need to identify and use appropriate ligands to quantify calcium in primary B-ALL samples. Also, an effort should be made to identify the calcium entry channel. For example, a potent and selective Orai inhibitor (CM-4620) could be used.

Minor aspects:

- Analysis of PLCg2 phosphorylation and induction of Ca⁺⁺ signaling should not only be shown in 1 or 2 cell lines but rather in a panel of >5 B-ALL cell lines to better depict the level of heterogeneity between different cell lines and mutational background. Likewise, the response to GC and Ca⁺⁺ inhibitors need to be extended to a larger panel of cell lines.

- In their introduction the authors emphasize the importance of PLCs and Ca⁺⁺ signaling in B-cells and B-lymphoid malignancies. However, PLCs and dependent Ca⁺⁺ signaling pathways have been described in other hematologic malignancies, such as AML. Related work should be cited and put in context with the results of this manuscript (e.g. PMID 34695195).

Reviewer #3:

Remarks to the Author:

The manuscript by Olivier Boyer and his colleagues addresses the issue of GC-resistance in B-ALL using a plethora of sophisticated molecular techniques and demonstrates key roles for CXCR4, PLCy2 and PKC in this process. They show that constitutively active PLCy2 is a characteristic feature of B-ALL, and that hyperphosphorylation of PLCy2 is induced by GC in resistant B-ALL cell lines. Downstream of these events, PKC becomes activated, eventually leading to calcium release. Hereby, GC promote an anti-apoptotic program and induce their own resistance. As previously reported, CXCR4 is highly expressed on acute lymphoblastic cells and represents a target of GC, thereby leading to the activation of the PLCy2/PKC axis. Overall, the experiments are technically sound and the conclusions well founded. However, the novelty of this work as well as the mechanistic insights are limited.

It has been known for many years that CXCR4 and phospholipase C are regulated by GC in different types of immune cells. Although these previous findings are central to the message of the manuscript, they are not discussed. Just to mention a few key publications that would have been informative for the reader to better place the results of the manuscript into the context of our current knowledge: Blood (1999) 93:2282, Blood (2009) 113:575, J Immunol (2004) 172:7154, J Allergy Clin Immunol (2009), 106:1132, Acta Neuropathol (2014) 127:713, Immunity (2018) 48:286.

The analysis of the energy metabolism falls a bit out of scope and this reviewer fails to understand how these experiments add to the understanding of GC-resistance in B-ALL. Furthermore, the processes of glucose transport and glycolysis are neglected.

The authors state in the Discussion section that the exact mechanism by which GC activate CXCR4 signaling has not yet been deciphered. This reviewer totally agrees with this view. Unfortunately,

the manuscript at hand does not advance our knowledge in this respect. Although triggering of CXCR4 signaling by GC is prominently placed in the title, the manuscript does not make any efforts to answer the question of how GC activate CXCR4. Is this achieved by genomic or non-genomic mechanisms, and is it a direct or rather an indirect effect? I would have definitely expected more insights into this central question. Just as an additional note, the authors seem to suggest that CXCR4 signaling was a consequence of its internalization. To this reviewer, however, it rather appears that CXCR4 directly signals to kinases such as ROCK, PI3K and ERK, and that the internalization of CXCR4 is just a secondary effect of its activation.

An important question arising from the conclusions made in this manuscript are their possible clinical consequences. Based on the observed overactivation of PLC γ 2 one could argue that using inhibitors of this enzyme is an option to treat B-ALL. However, there is no discussion of this possibility in the manuscript. In fact, it can be expected that such drugs would not be tolerable in patients. The addition of Plerixafor or another CXCR4 inhibitor would also be an option based on the findings of this manuscript. However, such an approach has already been published before. Randhawa et al. have shown in their article published in the *Br J Haematology* (2016) 174, 425 that inhibition of CXCR4 delays B-ALL progression and restores GC-sensitivity. With this result in mind, the novelty of the finding that CXCR4 plays a role in the context of B-ALL is strongly reduced. In addition, long-term treatment with Plerixafor is presumably not an option for patients too as revealed by the results of earlier clinical trials. In summary, the translatability of the results into the clinic is questionable.

Point-by-point answers to the reviewers' comments

Reviewer #1 - B-ALL therapy and resistance (Remarks to the Author):

“The manuscript by Abdoul-Azize et al. investigates PLC/CXCR4 mediated glucocorticoid resistance in B cell ALL. The authors show that dexamethasone treatment results in CXCR4 internalization, activation of PLC and results in therapy resistance. Treatment with PLC inhibitors or CXCR4 antagonists sensitize B ALL cells to dexamethasone. Overall, the manuscript is very dense and in some parts a challenge to follow. The manuscript would be significantly improved by increasing the clarity of the results section and figures. Additional scientific comments are below.”

Response: We thank the reviewer for his/her thorough assessment of our study, insightful summary and for his/her constructive comments. We have significantly modified the manuscript in order to augment the clarity of the results section and figures, as detailed below.

“The authors explore many different aspects of biology in this manuscript including cell cycle, transcriptional changes, mitochondrial biology, ROS production, Ca²⁺ signaling, and CXCR4. The addition of rescue experiments to show that the changes observed are critical to the mechanism being described would improve the study overall. For example, does antioxidant treatment rescue cell death showing ROS production is a key part of the mechanism?”

Response: We agree that experiments with antioxidant agents would be valuable but to reduce the density of the manuscript as requested and also following the comment of reviewer 3 (see comments below) that results on metabolism are out of scope, we rather propose to remove the following former results: Figure 6, Figure 8h-s, Extended Data Figure 10i, Supplementary Figure 8 and Supplementary Figure 9. Regarding rescue experiments, the manuscript already describes the rescue of CXCR4 agonist viability in B-ALL cell lines (Extended Data Figure 11c-f). Dexamethasone (Dex) rescues CXCR4 agonist mediated Ca²⁺ signaling (Figure 6v, 6w, Extended Data Figure 10a-d). In addition, we previously demonstrated that thapsigargin (intracellular Ca²⁺ inducer) rescues Dex-induced B-ALL cell death (Abdoul-Azize 2018, PMID 28423696).

“Are there any changes in PLC gamma, or CXCR4 expression from diagnosis to relapse in B ALL?”

Response: As requested, we explored the expression of PLC γ 2 and CXCR4 genes from diagnosis to relapse in publicly available B-ALL datasets. Analysis of paired data from two GEO-based expression datasets (PMID: 20072147, PMID: 21921043) showed no significant change (*two-tailed, Paired t test*) in PLC γ 2 and CXCR4 expression between diagnosis and relapse in B-ALL (see figure below), reinforcing our view that relapse/resistance is rather associated with activation of PLC pathway than a change in gene expression level. Accordingly, we added the following sentence in the Results section: “Also, analysis of paired data from publicly available datasets (ref. 46,47) showed no significant change in PLC γ 2 expression between diagnosis and relapse in B-ALL (data not shown)”.

“For combination treatments, it would be helpful to show the effect of vehicle, both single agents, and combination therapy to see if the effect of the combination treatment is greater than single agents alone. The same applies to the RNAi/CRISPR experiments. Showing control, siRNA or CRISPR cells with and without dexamethasone is essential to understand if the biology observed is due to CXCR4 knockdown/out alone or the loss of CXCR4 with dexamethasone.”

Response: We apologize for not showing these controls. Therefore, we have added them as follows:

- the effect of vehicles (Ctr) in Figure 5a,b,g; Figure 7b,g (former Figure 8b,e); Extended Data Figure 5a,c,d,f-i (former Extended Data Figure 4a,c,d,f-i); Extended Data Figure 6e-h (former Extended Data Figure 5e-h); Extended Data Figure 11i (former Extended Data Figure 9i), Supplementary Figure 7.
- siRNA control in Extended Data Figure 11k (former Extended Data Figure 9m).
- CRISPR controls in Figure 7c-f (former Figure 8c-d; Extended Data Figure 9k-l).

“Many of the experiments appear to have been performed with GC sensitive cell lines. While it is appreciated that some experiments were done in resistant lines, it would be helpful to repeat experiments in GC resistant lines (either intrinsically resistant or the authors Nalm6 dex-resistant line) to see if the mechanism is the same in dexamethasone resistant ALL. This would be particularly important to do for the mouse experiments.”

Response: We agree and have performed the suggested experiments by using two approaches. First, we made our Nalm6-LUC/GFP cells resistant to GC (method described in Extended Data Figure 2a). The new Extended Data Figure 12h shows the validation of the model in the NSG mouse model. We then tested this resistant line and showed similar results to the sensitive line (new Figure 8e-h). We also showed in this new experiment that PLC inhibition with U73122 improves mouse survival in the presence of Dex. Second, we used the intrinsically Dex-resistant

B-ALL cell line RCH-ACV (Karina A Kruth et al. Blood, 2017, PMID: 28424165) and showed similar results using CXCR4 and PLC inhibitors (new Figure 8i).

Reviewer #2 - Calcium signaling, PLC, leukaemia (Remarks to the Author):

“In their manuscript, Abdoul-Azize and colleagues describe glucocorticoid-triggered CXCR4 signaling as an activator of the PLC signalosome that may promote tumor resistance in B cell acute lymphoblastic leukemia. This is a timely and interesting topic and the authors have shown their expertise investigating signaling in hematologic neoplasms and immune responses. Here, they describe a relevant signaling axis that may contribute to resistance in B-ALL. While the exploration of the signaling pathway is detailed and conclusive, the manuscript falls short in relevant aspects that should be addressed to improve the quality of the already impressive manuscript.”

Response: We are delighted that this reviewer found our topic “timely and interesting” and that our results on Ca²⁺ signaling are “relevant” and “conclusive”. We have performed the requested additional experiments and respond point-by-point below.

“The authors indicate that PLCγ2 is highly expressed in B-ALL on the mRNA level. The mechanism behind this aberrant induction of PLCγ2 expression remains unexplained. Is the induction of PLCγ2 expression influenced by the underlying oncogene? Is the expression heterogeneous in samples with different driver mutations? The reason for high PLCγ2 expression specifically in B-ALL should be mechanistically explained, e.g. by using reporter systems for PLCγ2 in combination with genetic or pharmacologic screening approaches.”

Response: In Extended Data Figure 8, we showed that PLCγ2 gene expression was neither affected by pharmacological agents (AMD3100, AMD3465, WZ811 and MSK) nor by CXCR4 CRISPR knockout. We further explored the expression of PLCγ2 gene from diagnosis to relapse by searching publicly available B-ALL datasets and observed no difference. This point was also raised by reviewer 1 and we added the following sentence in the Results section: “Also, analysis of paired data from publicly available datasets (ref. 46,47) showed no significant change in PLCγ2 expression between diagnosis and relapse in B-ALL (data not shown)”. Additionally, we explored the expression of PLCγ2 gene according to gene mutations/fusions (ETV6-RUNX1, TCF3-PBX1, BCR-ABL1, KMT2A fusion) and NCI risk in publicly available B-ALL datasets (GSE181157, PMID: 34933343) and observed no difference (see figure below). Altogether, this reinforces our view that activation of PLCγ2 is more important than a change in its gene expression level in B-ALL.

“While the authors focus on PLCs as mediators of Ca^{2+} signaling and the role of PLC family members, they do not mention or investigate the role of PLC γ 2’s human homologue PLC γ 1. Although not highly expressed, it may have non-redundant functions in B-ALL (both on Ca^{2+} signaling and for activation of other signaling pathways). PLC γ 1 could partially compensate for its homologue. Inactivation of PLC γ 1 in B-ALL cell lines followed by functional analyses and investigation of Ca^{2+} signaling should be implemented as a relevant control. Also, specificity of PLC γ 2 inhibitors used (especially regarding the activity of PLC γ 1) need to be shown.”

Response: PLC pharmacological inhibitors are not specific to PLC γ 1 or PLC γ 2 but rather inhibit both. We agree with the reviewer that showing the effects of gene silencing of PLC γ 2 as well as PLC γ 1 would strengthen the manuscript. Therefore, we performed siRNA inhibition experiments and now show in new Extended Data Figure 4 that PLC γ 1 or PLC γ 2 silencing dampens Ca^{2+} signaling and leukemic cell viability in both sensitive and resistant cell lines.

“Functional analysis *in vivo* (Figure 8f) is restricted to NALM6-cells injected into mice. This data requires corroboration by the use of patient-derived xenografts. Several models of B-ALL xenografts are available and published. At least 2 different B-ALL xenografts (with different underlying genetics) should be for treatment with Dex, AMD3465 and the combination versus control.”

Response: In accordance with a similar comment from reviewer 1, we agree that additional *in vivo* data would strengthen the manuscript. For this, we performed new mouse experiments using two additional lines. First, using RCH-ACV cells, new data showed similar results with AMD3465 and after a Nalm-6 challenge (new Figure 8i). Second, we modified our Nalm6-LUC/GFP cells to render them resistant to GC (as shown in new Extended Data Figure 12h)

and tested them in the NSG mouse model. Results showed a similar effect of AMD3465 plus Dex combination as in the parental Nalm-6 (new Figure 8e-h). We further showed in these new experiments that PLC inhibition with U73122 also improved mouse survival.

“While the authors provide evidence for the activation of the CXCR4-PLC γ 2-Ca²⁺ signaling axis, functional perturbations to assess for further mechanistic exploration are lacking. Genetic screening in PLC γ 2 deficient (e.g. by pharmacologic or genetic inactivation) cell lines would further confirm the relevance of the proposed signaling axis and be helpful to identify relevant interactors (or unknown intermediates). This experiment could also be performed in GC resistant cell lines (versus sensitive) to validate the signaling axis.”

Response: This has now been performed in the new experiments described above.

“Investigation of Ca²⁺ signaling does not include relevant controls: Fura-2 measurements in cell lines should be controlled by a second assay. Intracellular Ca²⁺ sensors and assessment by microscopy could serve as an appropriate control. Likewise, it is not described in this manuscript whether CXCR4-PLC γ 2 signaling initiates a STIM and Orai dependent calcium signals. It has been demonstrated in primary and leukemic T and B lymphocytes, that hyperactivation of SOCE can drive a hyper-proliferative state. This also emphasizes the need to identify and use appropriate ligands to quantify calcium in primary B-ALL samples. Also, an effort should be made to identify the calcium entry channel. For example, a potent and selective Orai inhibitor (CM-4620) could be used.”

Response: Regarding the second assay, we confirmed that Dex induces Ca²⁺ influx by using Fluo-4 in addition to Fura-2, which yielded similar results (modified Supplementary Figure 4c and new Extended Data Figure 9). We also used Fura red in new Figure 8b. To further determine if STIM/Orai could affect GC sensitivity and Dex-mediated Ca²⁺ signaling in B-ALL cells, we used CM-4620 as requested plus three other pharmacological inhibitors (YM-58483, GSK-7975A and Synta66). We did not observe significant effects in combination with Dex (see figure below) as well as on Dex-mediated Ca²⁺ signaling, suggesting that other channels expressed in immune cells are involved, e.g. TRP family members. In this line, we previously showed that a TRPC3 inhibitor (Pyr3) improved Dex sensitivity in both primary and B-ALL cell line samples (Abdoul-azize *et al.*, PMID: 27179991). Additionally, we also showed that the TRP channel blocker SKF96365 enhanced Dex sensitivity in two B-ALL cell lines (Abdoul-azize *et al.*, PMID: 28423696). This is now indicated in the Results section: “Second, to determine whether STIM/Orai could affect GC sensitivity in B-ALL cells, we used several pharmacological inhibitors (CM-4620, YM-58483, GSK-7975A and Synta66) and did not observe significant effects in combination with Dex (data not shown), suggesting that other channels expressed in immune cells may be involved”.

“Minor aspects: Analysis of PLC γ 2 phosphorylation and induction of Ca²⁺ signaling should not only be shown in 1 or 2 cells lines but rather in a panel of >5 B-ALL cell lines to better depict the level of heterogeneity between different cell lines and mutational background. Likewise, the response to GC and Ca²⁺ inhibitors need to be extended to a larger panel of cell lines.”

Response: In addition to the 4 cell lines (Nalm-6, RS4;11, Reh, HAL-01) used in the submitted manuscript, we now provide new data using RCH-ACV cells (new Extended Data Figure 4 and new Extended Data Figure 9) in response to other comments and Ca²⁺ assessment in Nalm6 Dex-treated NSG mice *in vivo* (Figure 8b). Furthermore, we provide new *in vivo* data using RCH-ACV cells and also Dex-R Nalm-6 cells (new Figure 8e-i).

“Minor aspects: In their introduction the authors emphasize the importance of PLCs and Ca²⁺ signaling in B-cells and B-lymphoid malignancies. However, PLCs and dependent Ca²⁺ signaling pathways have been described in other hematologic malignancies, such as AML. Related work should be cited and put in context with the results of this manuscript (e.g. PMID 34695195).”

Response: This relevant reference (new ref. 21) has been added and the text has been modified accordingly: “Accordingly, PLC γ dysfunction is associated with a variety of immune disorders and cancers^{20,21}”.

Reviewer #3 - Glucocorticoid, mechanism-of-action, RNAseq (Remarks to the Author):

“The manuscript by Olivier Boyer and his colleagues addresses the issue of GC-resistance in B-ALL using a plethora of sophisticated molecular techniques and demonstrates key roles for CXCR4, PLC γ 2 and PKC in this process. They show that constitutively active PLC γ 2 is a characteristic feature of B-ALL, and that hyperphosphorylation of PLC γ 2 is induced by GC in resistant B-ALL cell lines. Downstream of these events, PKC becomes activated, eventually leading to calcium release. Hereby, GC promote an anti-apoptotic program and induce their own resistance. As previously reported, CXCR4 is highly expressed on acute lymphoblastic cells and represents a target of GC, thereby leading to the activation of the PLC γ 2/PKC axis. Overall, the experiments are technically sound and the conclusions well founded. However, the novelty of this work as well as the mechanistic insights are limited.”

Response: We thank the reviewer for considering that our experiments are “technically sound and the conclusions well founded”. We respond to comments below.

“It has been known for many years that CXCR4 and phospholipase C are regulated by GC in different types of immune cells. Although these previous findings are central to the message of the manuscript, they are not discussed. Just to mention a few key publications that would have been informative for the reader to better place the results of the manuscript into the context of our current knowledge: Blood (1999) 93:2282, Blood (2009) 113:575, J Immunol (2004) 172:7154, J Allergy Clin Immunol (2009), 106:1132, Acta Neuropathol (2014) 127:713, Immunity (2018) 48:286.”

Response: We have now included these relevant references in the revised manuscript and modified the text accordingly (see new ref. 61,62,63,64,68). We respectfully point out that we did not find the reference (J Allergy Clin Immunol 2009, 106:1132).

“The analysis of the energy metabolism falls a bit out of scope and this reviewer fails to understand how these experiments add to the understanding of GC-resistance in B-ALL. Furthermore, the processes of glucose transport and glycolysis are neglected.”

Response: We agree that our results on metabolism can be considered as beyond the scope of this article. Therefore, in accordance with reviewer 1 who also found the manuscript dense, we have removed the figures on energy and oxidative metabolism: Figure 6, Figure 8h-s, Extended Data Figure 10i, Supplementary Figure 8 and Supplementary Figure 9. The text has been simplified accordingly.

“The authors state in the Discussion section that the exact mechanism by which GC activate CXCR4 signaling has not yet been deciphered. This reviewer totally agrees with this view. Unfortunately, the manuscript at hand does not advance our knowledge in this respect. Although triggering of CXCR4 signaling by GC is prominently placed in the title, the manuscript does not make any efforts to answer the question of how GC activate CXCR4. Is this achieved by genomic or non-genomic mechanisms, and is it a direct or rather an indirect effect? I would have definitively expected more insights into this central question. Just as an additional note, the authors seem to suggest that CXCR4 signaling was a consequence of its internalization. To this reviewer, however, it rather appears that CXCR4 directly signals to kinases such as ROCK, PI3K and ERK, and that the internalization of CXCR4 is just a secondary effect of its activation.”

Response: To take into account the comment about the title, we have changed it to “Glucocorticoids paradoxically promote steroid resistance in B cell acute lymphoblastic leukemia through CXCR4/PLC signaling”.

Also, in the abstract, “Mechanistically, dexamethasone (Dex) provokes CXCR4 internalization, resulting in the activation of PLC-dependent Ca^{2+} and PKC signaling pathways” has been changed to “Mechanistically, dexamethasone (Dex) provokes CXCR4 signaling, resulting in the activation of PLC-dependent Ca^{2+} and PKC signaling pathways”

We agree that internalization of CXCR4 may be a secondary effect of its activation. To investigate this possibility, we determined whether Dex could induce Ca^{2+} signaling when CXCR4 internalization was prevented (*i.e.*, at $+4^\circ\text{C}$ rather than $+37^\circ\text{C}$). We found that in the absence of CXCR4 internalization, Ca^{2+} signaling was indeed triggered by Dex, confirming the reviewer’s view. This new result now appears in new Extended Data Figure 9 and the text has been modified accordingly.

Regarding the non-genomic vs genomic effects, the rapidity of observed effects of Dex (minutes) makes genomic mechanism very unlikely. To obtain further evidence, we performed an experiment by using a GC receptor antagonist (RU486) and showed that it had no effect on Dex-mediated Ca^{2+} signaling, confirming that the mechanism is non-genomic. This is now indicated in the discussion section: “The present experiments show nongenomic mechanisms at the membrane level as the GC receptor antagonist (RU486) had no effect on Dex-mediated Ca^{2+} signaling (data not shown)”. Furthermore, we added in the Discussion section the reference of a study demonstrating that GC can directly bind to GPCR proteins (Nature 2021; 589(7843):620-626, ref. 67).

“An important question arising from the conclusions made in this manuscript are their possible clinical consequences. Based on the observed overactivation of PLC γ 2 one could argue that using inhibitors of this enzyme is an option to treat B-ALL. However, there is no discussion of this possibility in the manuscript. In fact, it can be expected that such drugs would not be tolerable in patients.”

Response: We agree that using inhibitors of PLC γ 2 may represent a new option to treat B-ALL and this was not discussed in our manuscript. To provide experimental evidence supporting this view, we tested the effect of the inhibitor U73122 *in vivo* in mice. Data presented in new Figure 8e-i show that blocking PLC provides a therapeutic advantage. Accordingly, we have added a point in the Discussion section and cite a new reference (PMID 34695195) suggested by reviewer 2: “Based on the beneficial effects observed in mice, using PLC γ inhibitors may represent a treatment option in B-ALL. Although we did not observe overt toxicity in our mouse experiments or that PLC γ 1 invalidation does not impair normal hematopoietic stem cells²¹, the use of such pharmacological agents will have to be investigated with regard to their clinical benefit but also tolerability in patients”. Finally, in the abstract, we provide more detail about PLC inhibition required by the reviewer as follows: “Treatment with a CXCR4 antagonist or a PLC inhibitor improved survival of Dex-treated NSG mice *in vivo*”.

“The addition of Plerixafor or another CXCR4 inhibitor would also be an option based on the findings of this manuscript. However, such an approach has already been published before. Randhawa et al. have shown in their article published in the Br J Haematology (2016) 174, 425 that inhibition of CXCR4 delays B-ALL progression and restores GC-sensitivity. With this result in mind, the novelty of the finding that CXCR4 plays a role in the context of B-ALL is

strongly reduced. In addition, long-term treatment with Plerixafor is presumably not an option for patients too as revealed by the results of earlier clinical trials. In summary, the translatability of the results into the clinic is questionable.”

Response: We have already cited this reference (n°59 in the submitted version) to indicate that this work has shown that CXCR4 is a target to overcome stroma-mediated drug resistance in B-ALL. The novelty of our work is to demonstrate mechanistically that, independently of stroma, Dex promotes its own resistance through CXCR4/PLC, which has not been shown before. This is why we do not propose to use Plerixafor as long-term treatment of resistant B-ALL but as short-term and starting before administration of GC. The objective is to prevent resistance, which is a novel concept. We agree with this reviewer that this point could be made clearer in the manuscript. Therefore, we have modified the last sentence of the Discussion section: “These observations support further investigation into the use of CXCR4 inhibitors and/or PLC modulators as a rational preventive strategy to limit occurrence of resistance and improve the efficacy of Dex and chemotherapy in B-ALL patients”.

Reviewers' Comments:

Reviewer #1:

Remarks to the Author:

The authors have addressed all of my concerns and present here a significantly improved manuscript. Congratulations on a very nice study.

Reviewer #2:

Remarks to the Author:

The authors have prepared a thorough revision of their original manuscript and have given thoughtful consideration to the reviewer's comments.

This study brings novelty by providing mechanistic insight into a CXCR4/PLC signaling loop that promotes steroid resistance in B-ALL.

The modifications the authors have made to the manuscript have improved considerably an already compelling study.

The authors have addressed my original queries and I feel that this revised version is suitable for publication. This study will be an important addition to the field.

Reviewer #3:

Remarks to the Author:

I appreciate that the authors made strong efforts to constructively address all my concerns during the revision of their manuscript. Just as a short remark, the proposed reference that the authors could not identify contained a typo (J Allergy Clin Immunol 2000, 106:1132; Glucocorticoids preferentially upregulate functional CXCR4 expression in eosinophils). Altogether, the paragraph of the discussion section addressing the control of CXCR4 signaling by GC has been satisfactorily modified and the proposed new references have been cited. Nonetheless, I still suggest to additionally include the one mentioned above, too. As to the data on energy metabolism, I appreciate that the authors removed them, which clearly enhanced the focus of the manuscript. My concerns related to the mechanism of GC, namely the questions whether Dex induces CXCR4 internalization and whether GC signaling is genomic, were both well addressed in the title, abstract, text and by performing the experiment shown in the Extended Data Figure 9. However, I believe that the new study made to demonstrate the non-genomic nature of the control of Ca²⁺ signaling by Dex using RU486 should also be included in the Extended Data as I consider it very important. I would like to point out that my remark concerning the potential application of PLCγ inhibitors was nicely addressed by the new experiment shown in Figure 8, which truly represents a valuable addition to the manuscript. Lastly, I would like to thank the authors for clarifying the point that they consider Plerixafor application only as a potential short-term measure to treat B-ALL and as a supplementation to standard GC therapy. Collectively, I now recommend to accept the revised manuscript for publication.

Point-by-point answers to the reviewers' comments

Reviewer #1 - (Remarks to the Author):

“The authors have addressed all of my concerns and present here a significantly improved manuscript. Congratulations on a very nice study.”

Response: We thank the reviewer.

Reviewer #2 - (Remarks to the Author):

“The authors have prepared a thorough revision of their original manuscript and have given thoughtful consideration to the reviewer's comments.

This study brings novelty by providing mechanistic insight into a CXCR4/PLC signaling loop that promotes steroid resistance in B-ALL.

The modifications the authors have made to the manuscript have improved considerably an already compelling study.

The authors have addressed my original queries and I feel that this revised version is suitable for publication. This study will be an important addition to the field.”

Response: We thank the reviewer.

Reviewer #3 - (Remarks to the Author):

“I appreciate that the authors made strong efforts to constructively address all my concerns during the revision of their manuscript. Just as a short remark, the proposed reference that the authors could not identify contained a typo (J Allergy Clin Immunol 2000, 106:1132; Glucocorticoids preferentially upregulate functional CXCR4 expression in eosinophils). Altogether, the paragraph of the discussion section addressing the control of CXCR4 signaling by GC has been satisfactorily modified and the proposed new references have been cited. Nonetheless, I still suggest to additionally include the one mentioned above, too. As to the data on energy metabolism, I appreciate that the authors removed them, which clearly enhanced the focus of the manuscript. My concerns related to the mechanism of GC, namely the questions whether Dex induces CXCR4 internalization and whether GC signaling is genomic, were both well addressed in the title, abstract, text and by performing the experiment shown in the Extended Data Figure 9. However, I believe that the new study made to demonstrate the non-genomic nature of the control of Ca²⁺ signaling by Dex using RU486 should also be included in the Extended Data as I consider it very important. I would like to point out that my remark concerning the potential application of PLC γ inhibitors was nicely addressed by the new experiment shown in Figure 8, which truly represents a valuable addition to the manuscript. Lastly, I would like to thank the authors for clarifying the point that they consider Plerixafor application only as a potential short-term measure to treat B-ALL and as a supplementation to standard GC therapy. Collectively, I now recommend to accept the revised manuscript for publication.”

Response: We thank the reviewer for their careful comment. The proposed reference (ref. 64) as well as the effect of RU486 on Dex-mediated Ca²⁺ signaling (Supplementary Fig. 18f) have now been added.